# Structural conservation of antibiotic interaction with ribosomes

Helge Paternoga[1,5], Caillan Crowe-McAuliffe[1,5], Lars V. Bock ®[2], Timm O. Koller ®[1], Martino Morici ®[1], Bertrand Beckert[3], Alexander G. Myasnikov ®[3], Helmut Grubmüller ®[2], Jiří Nováček ®[4] & Daniel N. Wilson ®[1] ✉

The ribosome is a major target for clinically used antibiotics, but multidrug resistant pathogenic bacteria are making our current arsenal of antimicrobials obsolete. Here we present cryo-electron-microscopy structures of 17 distinct compounds from six different antibiotic classes bound to the bacterial ribosome at resolutions ranging from 1.6 to 2.2 Å. The improved resolution enables a precise description of antibiotic–ribosome interactions, encompassing solvent networks that mediate multiple additional interactions between the drugs and their target. Our results reveal a high structural conservation in the binding mode between antibiotics with the same scaffold, including ordered water molecules. Water molecules are visualized within the antibiotic binding sites that are preordered, become ordered in the presence of the drug and that are physically displaced on drug binding. Insight into RNA–ligand interactions will facilitate development of new antimicrobial agents, as well as other RNA-targeting therapies.

Extensive efforts over the past two decades have led to antibiotic–ribosome structures for every major class of clinically used ribosome-targeting antibiotic, including aminoglycosides, tetracyclines, lincosamides, macrolides, oxazolidinones, pleuromutilins, streptomycins and spectinomycins, thereby providing fundamental insight into their binding sites and mechanisms of action[1–6]. Generally, most of these antibiotic–ribosome structures are reported at resolutions ranging from 2.5 to 3.5 Å, with a few recent studies obtaining resolutions below 2.5 Å (refs. 7–11) and, to date, only one study better than 2 Å (ref. 12) (Supplementary Table 1). While comparison of the available antibiotic–ribosome structures reveals an overall similarity in terms of binding site for each class, in many cases there are profound differences evident with respect to the exact position and/or conformation of the modeled drugs, as well as in the surrounding ribosomal RNA that forms the drug binding site (Extended Data Fig. 1). These differences lead to divergent interaction networks being presented for antibiotics

with the same chemical scaffold (or even the exact same antibiotic) and most likely arise due to limitations in the resolution[13,14]. Therefore, high-resolution experimental data will be required to provide a more accurate description of the interactions of antibiotics with the ribosome. Additionally, higher resolution will also explain the extent to which ions and waters contribute to drug binding, and facilitate future structure-based design initiatives.

The role of water molecules in drug design for protein–ligand interactions has long been recognized, with multiple examples illustrating how water molecules can contribute to, or alternatively counteract, ligand binding and stability[15–19]. Moreover, hydration patterns within a protein can strongly influence the selectivity or promiscuity of the site for small molecules[15,19]. By contrast, comparatively little is known about the role of water in RNA–ligand interactions[20], partly due to the limited number of available high-resolution structures. Nevertheless, a recent study identified a single water-mediated interaction between

[1]Institute for Biochemistry and Molecular Biology, University of Hamburg, Hamburg, Germany. [2]Theoretical and Computational Biophysics Department, Max Planck Institute for Multidisciplinary Sciences, Göttingen, Germany. [3]Dubochet Center for Imaging at EPFL-UNIL, Batiment Cubotron, Lausanne, Switzerland. [4]Central European Institute of Technology (CEITEC), Masaryk University, Brno, Czech Republic. [5]These authors contributed equally: Helge Paternoga, Caillan Crowe-McAuliffe. ✉e-mail: Daniel.Wilson@chemie.uni-hamburg.de

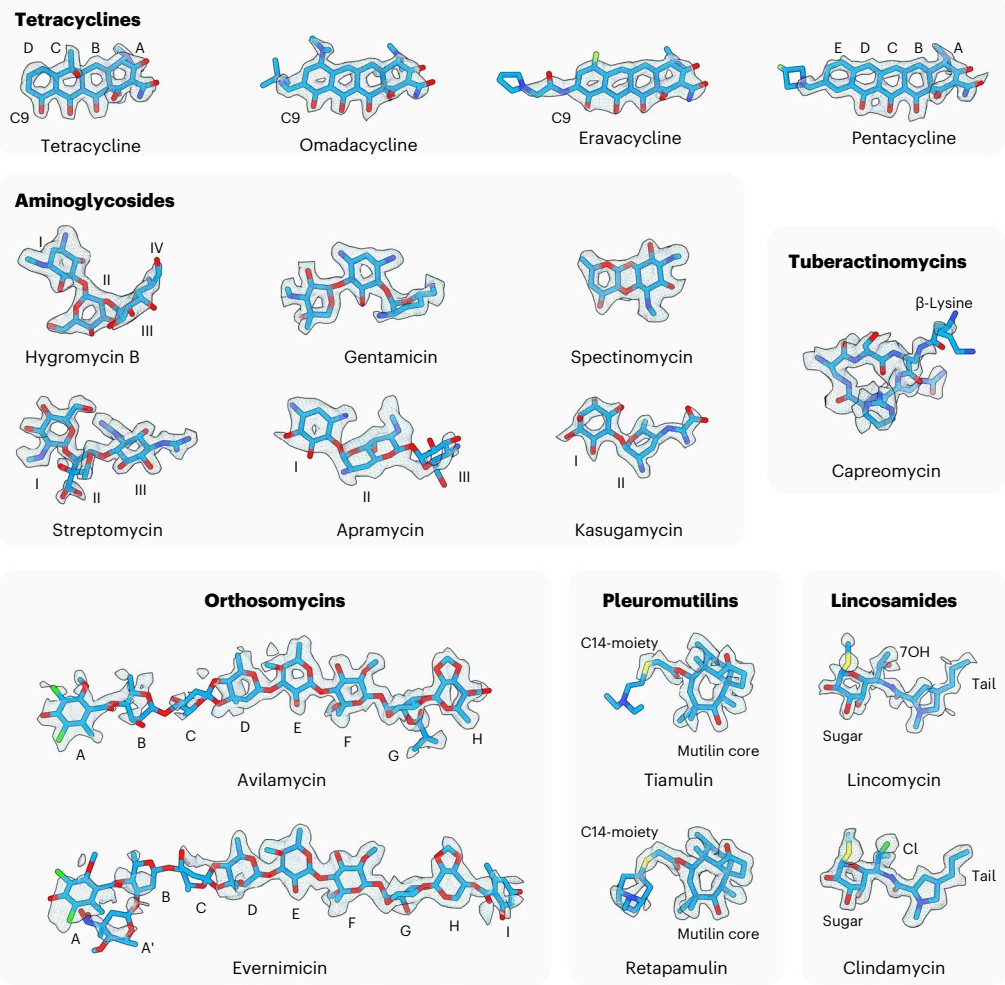

**Fig. 1 | Cryo-EM maps and models for 17 ribosome-targeting antibiotics.**
Segmented cryo-EM map densities (transparent gray) and molecule models
(colored by atom) are shown for tetracyclines (tetracycline, omadacycline,
eravacycline and pentacycline), aminoglycosides (hygromycin B, gentamicin,
spectinomycin, streptomycin, apramycin and kasugamycin), tuberactinomycin
(capreomycin), orthosomycin (avilamycin and evernimicin), pleuromutilin
(tiamulin and retapamulin) and lincosamide (lincomycin and clindamycin)
antibiotics.

the macrolide antibiotic erythromycin and the ribosome that is critical
for drug binding and inhibition[21]. This indicates that solvent-mediated
interactions can play a critical role for antibiotic binding to ribosomes,
and highlights the potential importance of water for other therapeuti-
cally relevant ligand–RNA interactions[22–24]. Here, we provide an atlas of
high-resolution antibiotic–ribosome interactions for several clinically
used classes of antibiotics. To do this, we have determined structures
of 17 different compounds in complex with the ribosome at 1.6–2.2 Å
resolution, which has allowed us to precisely describe the contacts of
the compounds within their binding pockets and has revealed a high
structural conservation of the interaction networks between related
families of compounds.

## Results
### Structures of antibiotic–ribosome complexes
To achieve maximum throughput, we determined structures with
multiple antibiotics in complex with the same ribosome. To ensure
homogeneity of the antibiotic–ribosome complexes, which is impor-
tant for obtaining high resolution, we used highly purified in vitro
reassociated *Escherichia coli* 70S ribosomes, to which each cocktail
of antibiotics was added. We generated five distinct antibiotic–
ribosome complexes and subjected them to single particle
cryo-electron-microscopy (cryo-EM) analysis (Methods). This study

visualizes antibiotics targeting both the small ribosomal subunit
(SSU), including tetracycline, a pentacycline and the third genera-
tion glycylcyclines omadacycline and eravacycline, as well as the
tuberactinomycin capreomycin, the aminocyclitol spectinomycin
and the aminoglycosides streptomycin, kasugamycin, gentamicin
and hygromycin B (Fig. 1). Large ribosomal subunit (LSU) antibiotics
visualized include the orthosomycins avilamycin and evernimicin, the
pleuromutilins tiamulin and retapamulin, as well as the lincosamides
lincomycin and clindamycin (Fig. 1). Cryo-EM data were collected on
Titan Krios transmission electron microscopes (TEMs) using direct
electron detectors and processed with RELION[25] (Methods). After 3D
refinement, the 70S ribosomes from the five datasets displayed average
resolutions of 1.8–2.0 Å. The 70S reconstructions were then subjected
to focused refinement, yielding average resolutions of 1.8–2.2 Å for
the SSU body and head, and 1.6–2.0 Å for the LSU core (Extended Data
Figs. 2–4 and Tables 1–3). The improved resolution of the structures
enabled us to generate and refine high-quality molecular models, with
excellent validation parameters, including very low clash (0.4–0.9) and
MolProbity scores (0.7–1.0) (Tables 1–3).

Inspection of the cryo-EM maps revealed densities for 17 com-
pounds bound within their primary binding sites (Fig. 1): specifically,
ten on the SSU (tetracycline, omadacycline, eravacycline, pentacycline,
spectinomycin, streptomycin, apramycin, kasugamycin, gentamicin

**Table 1 | Cryo-EM data collection, refinement and validation statistics for SSU head**

| | Dataset 1 (EMDB-16520) (PDB 8CA7) | Dataset 2 (EMDB-16536) (PDB 8CAZ) | Dataset 3 (EMDB-16615) (PDB 8CF1) | Dataset 4 (EMDB-16620) (PDB 8CF8) | Dataset 5 (EMDB-16644) (PDB 8CGI) |
|---|---|---|---|---|---|
| **Data collection and processing** | | | | | |
| Magnification | ×165,000 | ×270,000 | ×270,000 | ×270,000 | ×105,000 |
| Acceleration voltage (kV) | 300 | 300 | 300 | 300 | 300 |
| Electron exposure ($e^-/Å^2$) | 11 | 6 | 6 | 6 | 25 |
| Defocus range (μm) | −0.4 to −1.6 | −0.4 to −1.0 | −0.4 to −1.0 | −0.4 to −1.0 | −0.4 to −1.6 |
| Pixel size (Å) | 0.53 | 0.45 | 0.45 | 0.45 | 0.51 |
| Symmetry imposed | C1 | C1 | C1 | C1 | C1 |
| Initial particle images (no.) | 754,663 | 219,953 | 757,044 | 464,723 | 2,146,827 |
| Final particle images (no.) | 514,855 | 179,724 | 419,159 | 275,137 | 1,301,160 |
| Map resolution (Å) | 2.06 | 2.11 | 1.82 | 2.20 | 1.89 |
| FSC threshold | 0.143 | 0.143 | 0.143 | 0.143 | 0.143 |
| **Refinement** | | | | | |
| Initial model used (PDB code) | 7K00 | 7K00 | 7K00 | 7K00 | 7K00 |
| Model resolution (masked, Å) | 2.1 | 2.1 | 1.8 | 2.2 | 1.9 |
| FSC threshold | 0.5 | 0.5 | 0.5 | 0.5 | 0.5 |
| CC (mask) | 0.75 | 0.86 | 0.85 | 0.90 | 0.82 |
| CC (volume) | 0.74 | 0.85 | 0.83 | 0.89 | 0.81 |
| Map sharpening $B$ factor ($Å^2$) | −33.3 | −26.7 | −49.3 | −15.4 | −39.2 |
| Model composition | | | | | |
| Nonhydrogen atoms | 18,093 | 19,193 | 18,957 | 19,324 | 17,553 |
| Protein residues | 848 | 1,095 | 988 | 1,102 | 841 |
| RNA residues | 502 | 460 | 485 | 460 | 460 |
| Waters | 543 | 618 | 677 | 681 | 948 |
| Magnesium (MG) | 26 | 29 | 31 | 32 | 32 |
| Potassium (K) | 14 | 13 | 12 | 12 | 10 |
| Antibiotics* | SCM, U3B | – | TAC | YQM | P8F |
| $B$ factors ($Å^2$) | | | | | |
| Protein | 61.72 | 67.40 | 53.15 | 74.43 | 56.72 |
| RNA | 59.78 | 46.08 | 43.19 | 46.74 | 50.94 |
| Ligand | 37.73 | 32.92 | 27.78 | 39.57 | 39.21 |
| Water | 38.93 | 35.62 | 29.73 | 37.05 | 41.92 |
| R.m.s. deviations | | | | | |
| Bond lengths (Å) | 0.008 | 0.008 | 0.008 | 0.008 | 0.008 |
| Bond angles (°) | 1.520 | 1.521 | 1.501 | 1.542 | 1.577 |
| **Validation** | | | | | |
| MolProbity score | 0.89 | 0.89 | 0.90 | 1.04 | 0.78 |
| Clashscore | 0.60 | 0.52 | 0.76 | 0.76 | 0.63 |
| Poor rotamers (%) | 0.71 | 0.55 | 0.49 | 0.98 | 0.71 |
| Ramachandran plot | | | | | |
| Favored (%) | 96.84 | 96.63 | 97.08 | 95.76 | 97.69 |
| Allowed (%) | 3.03 | 3.27 | 2.81 | 4.06 | 2.19 |
| Disallowed (%) | 0.12 | 0.09 | 0.10 | 0.18 | 0.12 |
| Ramachandran $Z$ score | −1.46 | −1.41 | −0.76 | −1.40 | −0.30 |

[a]SCM (spectinomycin), U3B (omadacycline), TAC (tetracycline), YQM (eravacycline), P8F (pentacycline)

and hygromycin B), six on the LSU (avilamycin, evernimicin, lincomycin, clindamycin, tiamulin and retapamulin) and capreomycin that binds at the interface between the SSU and LSU. While the densities for the core-scaffold of each antibiotic were very well resolved and could be modeled unambiguously, flexibility was evident for parts of some molecules: for example, the C9-moieties of omadacycline, eravacycline

**Table 2 | Cryo-EM data collection, refinement and validation statistics for SSU body**

| | Dataset 1 (EMDB-16526) (PDB 8CAI) | Dataset 2 (EMDB-16612) (PDB 8CEP) | Dataset 3 (EMDB-16645) (PDB 8CGJ) | Dataset 4 (EMDB-16650) (PDB 8CGR) | Dataset 5 (EMDB-16651) (PDB 8CGU) |
|---|---|---|---|---|---|
| **Data collection and processing** | | | | | |
| Magnification | ×165,000 | ×270,000 | ×270,000 | ×270,000 | ×105,000 |
| Acceleration voltage (kV) | 300 | 300 | 300 | 300 | 300 |
| Electron exposure (e⁻/Å²) | 11 | 6 | 6 | 6 | 25 |
| Defocus range (μm) | −0.4 to 1.6 | −0.4 to −1.0 | −0.4 to −1.0 | −0.4 to −1.0 | −0.4 to −1.6 |
| Pixel size (Å) | 0.53 | 0.45 | 0.45 | 0.45 | 0.51 |
| Symmetry imposed | C1 | C1 | C1 | C1 | C1 |
| Initial particle images (no.) | 754,663 | 219,953 | 757,044 | 464,723 | 2,146,827 |
| Final particle images (no.) | 514,855 | 179,724 | 419,159 | 275,137 | 1,301,160 |
| Map resolution (Å) | 2.08 | 2.04 | 1.79 | 2.12 | 1.89 |
| FSC threshold | 0.143 | 0.143 | 0.143 | 0.143 | 0.143 |
| **Refinement** | | | | | |
| Initial model used (PDB code) | 7K00 | 7K00 | 7K00 | 7K00 | 7K00 |
| Model resolution (masked, Å) | 2.1 | 2.0 | 1.8 | 2.1 | 1.9 |
| FSC threshold | 0.5 | 0.5 | 0.5 | 0.5 | 0.5 |
| CC (mask) | 0.81 | 0.85 | 0.83 | 0.88 | 0.81 |
| CC (volume) | 0.80 | 0.84 | 0.81 | 0.87 | 0.81 |
| Map sharpening $B$ factor (Å²) | -32.2 | −24.5 | −53.9 | −5.1 | −36.6 |
| Model composition | | | | | |
| Nonhydrogen atoms | 37,814 | 35,490 | 38,708 | 35,552 | 35,196 |
| Protein residues | 1,552 | 1,279 | 1,523 | 1,272 | 1,266 |
| RNA residues | 1,114 | 1,077 | 1,161 | 1,090 | 1,071 |
| Waters | 1,416 | 1,933 | 1,616 | 1,928 | 1,869 |
| Magnesium (MG) | 55 | 54 | 63 | 57 | 58 |
| Potassium (K) | 29 | 25 | 25 | 28 | 24 |
| Antibiotics* | SCM, HYO (3), 5IO | KSG, CA7 (5) | TAC, 5IO | AM2 (3) | LLL (4) |
| $B$ factors (Å²) | | | | | |
| Protein | 82.43 | 53.52 | 59.68 | 55.00 | 60.73 |
| RNA | 54.40 | 40.44 | 45.74 | 48.01 | 48.21 |
| Ligand | 55.57 | 66.05 | 30.78 | 54.19 | 52.66 |
| Water | 49.17 | 39.57 | 32.33 | 42.61 | 46.19 |
| R.m.s. deviations | | | | | |
| Bond lengths (Å) | 0.008 | 0.008 | 0.008 | 0.007 | 0.008 |
| Bond angles (°) | 1.496 | 1.536 | 1.515 | 1.585 | 1.581 |
| **Validation** | | | | | |
| MolProbity score | 0.95 | 0.72 | 0.87 | 0.87 | 0.71 |
| Clashscore | 0.59 | 0.43 | 0.37 | 0.93 | 0.38 |
| Poor rotamers (%) | 0.86 | 0.66 | 0.40 | 0.10 | 0.38 |
| Ramachandran plot | | | | | |
| Favored (%) | 96.22 | 97.68 | 96.41 | 97.58 | 97.65 |
| Allowed (%) | 3.71 | 2.32 | 3.46 | 2.42 | 2.35 |
| Disallowed (%) | 0.07 | 0.00 | 0.14 | 0.00 | 0.00 |
| Ramachandran $Z$ score | −1.66 | −0.88 | −1.33 | −0.51 | −0.61 |

ᵃSCM (spectinomycin), HYO (hygromycin B), 5IO (streptomycin), KSG (kasugamycin), CA7 (capreomycin), TAC (tetracycline), AM2 (apramycin), LLL (gentamicin).

and the pentacycline, ring III of apramycin, the β-lysine of capreomycin and the C14-moieties of tiamulin and retapamulin (Fig. 1). With respect to the orthosomycins, the central rings D–H of evernimicin and avilamycin were well resolved (Fig. 1), whereas the other rings exhibited flexibility, especially rings B and C for Evn and/or Avn and rings A′ and I for Evn, presumably because these moieties do not interact directly with the

**Table 3 | Cryo-EM data collection, refinement and validation statistics for LSU**

| | Dataset 1 (EMDB-16530) (PDB 8CAM) | Dataset 2 (EMDB-16613) (PDB 8CEU) | Dataset 3 (EMDB-16646) (PDB 8CGK) | Dataset 4 (EMDB-16641) (PDB 8CGD) | Dataset 5 (EMDB-16652) (PDB 8CGV) |
|---|---|---|---|---|---|
| **Data collection and processing** | | | | | |
| Magnification | ×165,000 | ×270,000 | ×270,000 | ×270,000 | ×105,000 |
| Acceleration voltage (kV) | 300 | 300 | 300 | 300 | 300 |
| Electron exposure ($e^-/Å^2$) | 11 | 6 | 6 | 6 | 25 |
| Defocus range (μm) | −0.4 to −1.6 | −0.4 to −1.0 | −0.4 to 1.0 | −0.4 to 1.0 | −0.4 to −1.6 |
| Pixel size (Å) | 0.53 | 0.45 | 0.45 | 0.45 | 0.51 |
| Symmetry imposed | C1 | C1 | C1 | C1 | C1 |
| Initial particle images (no.) | 754,663 | 219,953 | 757,044 | 464,723 | 2,146,827 |
| Final particle images (no.) | 514,855 | 179,724 | 419,159 | 275,137 | 1,301,160 |
| Map resolution (Å) | 1.86 | 1.83 | 1.64 | 1.98 | 1.65 |
| FSC threshold | 0.143 | 0.143 | 0.143 | 0.143 | 0.143 |
| **Refinement** | | | | | |
| Initial model used (PDB code) | 7K00 | 7K00 | 7K00 | 7K00 | 7K00 |
| Model resolution (masked, Å) | 1.9 | 1.8 | 1.6 | 2.0 | 1.7 |
| FSC threshold | 0.5 | 0.5 | 0.5 | 0.5 | 0.5 |
| CC (mask) | 0.76 | 0.85 | 0.80 | 0.84 | 0.77 |
| CC (volume) | 0.76 | 0.84 | 0.78 | 0.82 | 0.76 |
| Map sharpening $B$ factor ($Å^2$) | −26.4 | −18.0 | −32.0 | −22.0 | −27.0 |
| Model composition | | | | | |
| Nonhydrogen atoms | 86,648 | 92,830 | 88,150 | 94,181 | 87,899 |
| Protein residues | 2,767 | 3,028 | 2,792 | 3,059 | 2,819 |
| RNA residues | 2,724 | 2,872 | 2,738 | 2,867 | 2,743 |
| Waters | 6,113 | 6,830 | 7,101 | 8,357 | 6,570 |
| Magnesium (MG) | 246 | 220 | 260 | 254 | 219 |
| Potassium (K) | 72 | 81 | 81 | 85 | 83 |
| Antibiotics* | 6O1 | G34, CA7 (8) | 3QB, 6UQ | CLY, AM2 | MUL, P8F (2) |
| $B$ factors ($Å^2$) | | | | | |
| Protein | 52.68 | 48.60 | 36.22 | 50.57 | 43.14 |
| RNA | 57.51 | 44.15 | 41.64 | 45.61 | 40.17 |
| Ligand | 51.25 | 58.66 | 23.33 | 37.21 | 31.73 |
| Water | 41.80 | 38.02 | 30.76 | 42.29 | 36.51 |
| R.m.s. deviations | | | | | |
| Bond lengths (Å) | 0.009 | 0.008 | 0.008 | 0.008 | 0.008 |
| Bond angles (°) | 1.547 | 1.556 | 1.531 | 1.579 | 1.585 |
| **Validation** | | | | | |
| MolProbity score | 0.90 | 0.82 | 0.67 | 0.85 | 0.74 |
| Clashscore | 0.61 | 0.53 | 0.37 | 0.70 | 0.52 |
| Poor rotamers (%) | 0.27 | 0.65 | 0.26 | 0.36 | 0.35 |
| Ramachandran plot | | | | | |
| Favored (%) | 96.84 | 97.22 | 97.84 | 97.36 | 97.71 |
| Allowed (%) | 3.05 | 2.74 | 2.16 | 2.58 | 2.29 |
| Disallowed (%) | 0.11 | 0.03 | 0.00 | 0.07 | 0.00 |
| Ramachandran $Z$ score | −1.28 | −1.11 | −0.57 | −0.73 | −0.49 |

[a]6O1 (evernimicin), G34 (retapamulin), CA7 (capreomycin), 3QB (lincomycin), 6UQ (avilamycin), CLY (clindamycin), AM2 (apramycin), MUL (tiamulin), P8F (pentacycline).

ribosome. Overall, the binding sites observed here are fundamentally similar to those observed previously and therefore consistent with their proposed mechanisms of action to inhibit protein synthesis[1–6].

Similar to previous studies[7,26–29], we also observed secondary binding sites for a number of antibiotics that is likely due to the high antibiotic concentrations used (Extended Data Fig. 5), but not the

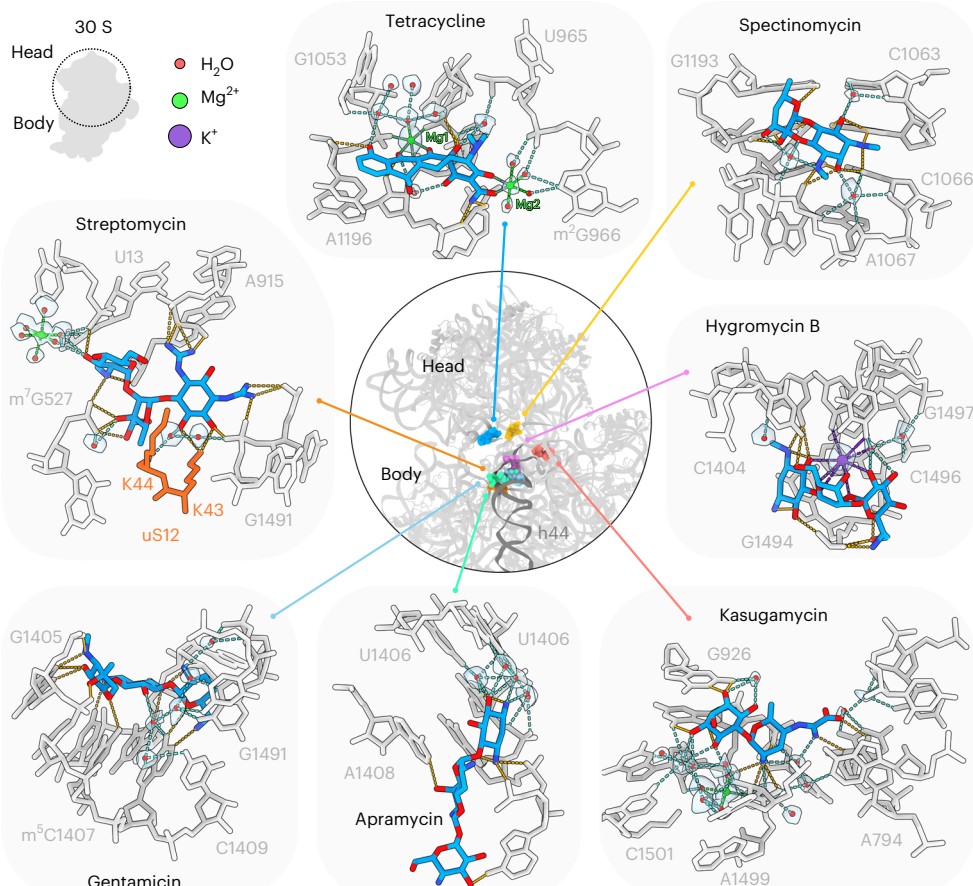

**Fig. 2 | Structures of antibiotics targeting the SSU.** The central ring shows a superimposition of the binding sites on the SSU (gray) of the antibiotics tetracycline (blue), spectinomycin (yellow), hygromycin B (pink), kasugamycin (red), apramycin (green), gentamicin (cyan) and streptomycin (orange), which is surrounded by insets highlighting the interactions between the drug and the 16S rRNA (gray), waters (red spheres with gray transparent density), magnesium ions (green spheres), putative K⁺ ions (purple sphere with transparent gray density) and uS12 (orange). Potential hydrogen bonds are indicated as dashed lines, colored orange for direct interaction between the drug and the small subunit, cyan for water-mediated interactions, green for Mg²⁺ ion coordination and purple for K⁺ coordination.

secondary site for kasugamycin observed previously[30]. Only one secondary binding site was observed for tetracycline, namely, on the SSU, overlapping the previously reported Tet2 site[26], but with an inverted orientation (Extended Data Fig. 5a). No secondary sites for other tetracycline derivatives were observed on the SSU, although two sites on the LSU were observed for eravacycline and pentacycline (Extended Data Fig. 5b), as reported previously for eravacycline[31]. Two secondary sites for apramycin were observed on the SSU, overlapping previously reported sites[7,29], as well as one new site on the LSU (Extended Data Fig. 5c). Of the two secondary sites observed for gentamicin on the SSU, one was reported previously[28] (Extended Data Fig. 5d), whereas we identify two new secondary sites on the SSU for hygromycin B (Extended Data Fig. 5e). We observed nine secondary binding sites for capreomycin, two on the SSU, two at the SSU–LSU interface and five on the LSU (Extended Data Fig. 5f). Although secondary sites for capreomycin were not observed previously[32], four sites at similar (but distinct) locations within H69 on the LSU were reported for the related tuberactinomycin viomycin[33] (Extended Data Fig. 5f). While the secondary binding sites are not likely to be physiological relevant, they nevertheless provide a wealth of additional information on small molecule–RNA interactions.

**Direct and indirect ribosomal interaction of antibiotics**

The level of detail of the antibiotic–ribosome complexes determined here enables a more accurate description of the interaction of each class of antibiotics on the SSU (Fig. 2 and Supplementary Videos 1–7)

and LSU (Fig. 3 and Supplementary Videos 8–11). This encompasses hydrogen bond interactions, either directly with the ribosomal components, or indirectly via ion- or water-mediated contacts (Figs. 2 and 3, Supplementary Videos 1–11 and Supplementary Figs. 1–17). Generally, we observed that the compounds studied here use between 10 and 20 hydrogen bonds to interact with the ribosome (Figs. 2 and 3), the exception being the pleuromutilins, such as tiamulin, where only six hydrogen bonds are possible: four direct and two water-mediated (Fig. 3). By contrast, the orthosomycin avilamycin establishes 18 hydrogen bond interactions (15 direct and three water-mediated) with the ribosome (Fig. 3), which correlates with the large size (8–10 sugar rings) and highly polar nature (comprising 32–38 oxygens) of these compounds (Fig. 1). However, large size is not a prerequisite to establish so many interactions, since the much smaller aminoglycoside kasugamycin (two rings comprising nine oxygens and three nitrogens) can form 20 hydrogen bonds (ten direct and ten water-mediated) with the ribosome (Fig. 2). Generally, the antibiotics form direct hydrogen bonds with a mixture of both the backbone (ribose- and phosphate-oxygens) and nucleobase of the rRNA. However, the ratio between backbone and nucleobase interactions varies, ranging from 31% (four from 13) for avilamycin (Fig. 3) to 83% (10 from 12) for spectinomycin (Fig. 2). The exceptions are (1) streptomycin, which does not make any nucleobase-specific interactions, but instead forms 14 hydrogen bonds with the rRNA backbone of nucleotides located in helices h2 and h18 (Fig. 2), and (2) tetracycline, which forms at least six direct hydrogen bonds to the backbone of

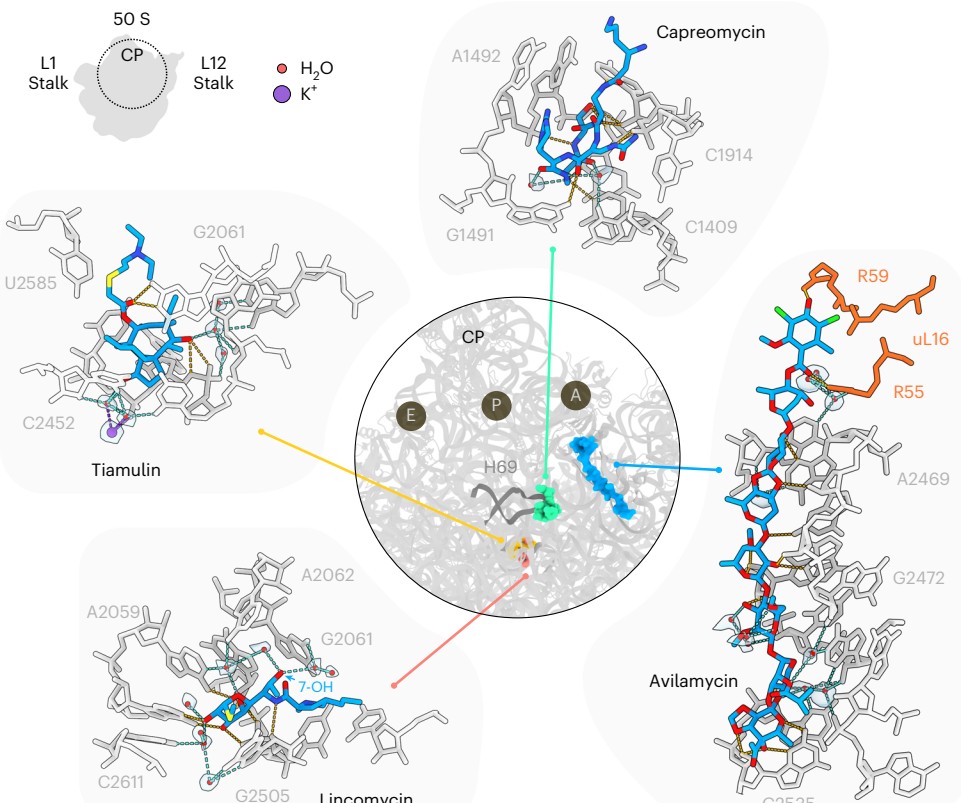

**Fig. 3 | Structures of antibiotics targeting the interface and LSU.** The central ring shows a superimposition of the binding sites on the LSU (gray) of the antibiotics capreomycin (green), avilamycin (blue), lincomycin (red) and tiamulin (yellow), which is surrounded by insets highlighting the interactions between the drug and the rRNA (gray), waters (red spheres with gray transparent density), putative $K^+$ ions (purple sphere with transparent gray density) and uL16 (orange). Potential hydrogen bonds are indicated as dashed lines, colored orange for direct interaction between the drug and the small subunit, cyan for water-mediated interactions, green for $Mg^{2+}$ ion coordination and purple for $K^+$ coordination.

nucleotides in h34, and, as noted previously[26], the only sequence specific interaction is a stacking interaction with C1054 (Fig. 2), thus explaining the broad spectrum of activity of these classes[4,34].

In addition to direct hydrogen bond interactions, we also observed indirect interactions mediated via magnesium ions for tetracycline and a putative potassium ion for hygromycin B (Fig. 2). For tetracycline, the primary magnesium ion (Mg1) is fully coordinated by six oxygen atoms, three from the nucleotides within h33 of the 16S rRNA, two from rings B and C of tetracycline and one from a well-defined water molecule (Fig. 2). Compared to previous structures, we could more precisely model the coordination extent, geometry and distances between the Mg1 and the oxygen atoms (Extended Data Fig. 6). The secondary magnesium ion (Mg2) is also coordinated by six oxygen atoms, two from ring A of tetracycline and four from water molecules, which are within hydrogen bonding distance to U965 within h31 of the 16S rRNA (Fig. 2). This differs significantly from the less defined Mg2 coordination observed in previous studies (Extended Data Fig. 6). Unlike Mg2 that is only observed on tetracycline binding, Mg1 is also present in the absence of the drug[12] (Extended Data Fig. 7). This indicates that on tetracycline binding, two waters that coordinate Mg1 are displaced, but full coordination of the Mg1 is restored by substituting the waters with the oxygens from ring B and C of tetracycline. For hygromycin B, the conformation observed in our structure does not permit coordination of a $Mg^{2+}$ ion as modeled in the eukaryotic 80S-hygromycin B structure[35]. However, density is observed for a putative $K^+$ ion that is coordinated by rings I, II and IV of hygromycin B, 16S rRNA nucleotide G1497 and a water molecule (Fig. 2). The $K^+$ ion was not reported previously[35–37], and is not observed in the absence of the drug,

suggesting that it is stabilized on drug binding. We also observed a putative $K^+$ ion involved in mediating similar interactions of hygromycin B at a secondary binding site located at the junction between h22 and h23 of the 16S rRNA (Extended Data Fig. 5e).

For each antibiotic analyzed in this study, we observed multiple water molecules that mediate interactions between the drugs and the ribosomal components (Figs. 2 and 3). The extent of solvation varies considerably depending on the antibiotic class, such that only a few water molecules are involved in the binding of, for example, avilamycin, capreomycin, hygromycin B and tiamulin, whereas multiple waters are observed for others, as exemplified by tetracycline and kasugamycin (Figs. 2 and 3). While most water molecules are coordinated by oxygen atoms within the drugs, coordination by nitrogen atoms is also observed, for example by gentamicin and kasugamycin (Fig. 2) as well as capreomycin (Fig. 3). We note that, with the exception of capreomycin, there is a high prevalence of oxygen over nitrogen atoms in ribosome-targeting antibiotics, which likely explains the preference for oxygen-mediated interactions. Generally, oxygen atoms within the rRNA also make the highest contribution to coordinating waters that mediate drug interactions, being present both in the nucleobases as well as in the backbone of the rRNA. This is exemplified by spectinomycin, where two waters are coordinated by the oxygens (O2) located in the nucleobases of C1063 and C1066, together with their respective ribose 2′ oxygens (Fig. 2). For tetracycline and streptomycin, all the water molecules mediating direct drug interactions are coordinated by backbone phosphate-oxygens and/or ribose oxygens, including in the case of streptomycin the backbone oxygen of Lys44 of ribosomal protein uS12 (Fig. 2). Exceptions include two water

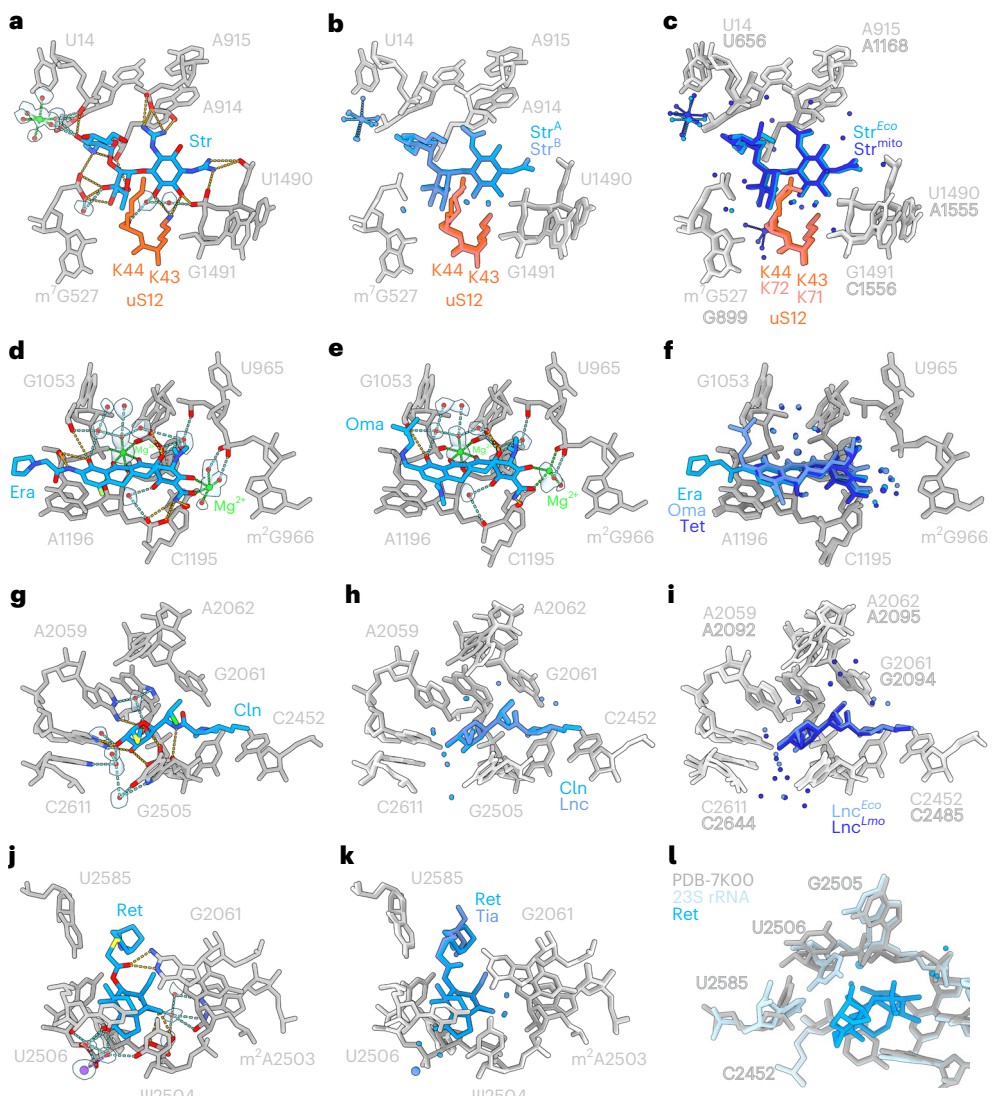

**Fig. 4 | Structural conservation of antibiotic binding to ribosomes.**
**a**, Interaction of streptomycin (Str[A]) on the SSU at 2.0 Å. **b**, Superimposition of streptomycin (Str[A]) from **a** with streptomcyin (Str[B]) determined at 1.8 Å (from Fig. 2). **c**, Superimposition of streptomycin determined here on *E. coli* (Str[Eco]) at 1.8 Å (Str[B] from Fig. 2) with streptomycin on the human mitochondrial SSU body (Str[mito]) at 2.23 Å (ref. 11). **d**,**e**, Interaction of eravacycline (Era) (**d**) and omadacycline (**e**) on the SSU at 2.1 and 2.2 Å, respectively. **f**, Superimposition of eravacycline (Era) from **d**, omadacycline from (and tetracycline (Tet) (from Fig. 2).

**g**, Interaction of clindamycin (Cln) on the LSU at 2.0 Å. **h**, Superimposition of clindamycin (Cln) from **g** with lincomycin (Lnc) determined at 1.6 Å (from Fig. 3). **i**, Superimposition of lincomycin determined here on *E. coli* (Lnc[Eco]) at 1.8 Å (from Fig. 2) with lincomycin on the *L. monocytogenes* 70S ribosome (Lnc[Lm.] at 2.1 Å, ref. 47). **j**, Interaction of retapamulin (Ret) on the LSU at 1.9 Å. **k**, Superimposition of retapamulin (Ret) from **j** with tiamulin (Tia) (from Fig. 3). **l**, Superimposition of Ret (blue) and 23S rRNA (cyan) from Ret-LSU structure with *E. coli* 70S ribosome lacking any drug in the A-site of the PTC (gray, PDB ID 7K00)[12].

molecules coordinated by ring III of streptomycin that are involved in the octahedral coordination of a Mg[2+] ion (Fig. 2), which is also observed in the absence of streptomycin[12]. The O6 of ring I of kasugamycin also interacts with a water that coordinates a Mg[2+] ion (Fig. 2), which is also present in the absence of the drug[12]. Nevertheless, nitrogen atoms present in nucleobases are also observed to coordinate water molecules, as exemplified by lincomycin, where five nitrogen atoms in A2059, G2061, A2062, A2503 and C2611 participate in the coordination of four distinct water molecules (Fig. 3). In addition to the first layer of water molecules that directly mediate antibiotic–ribosome interactions, we also observed the appearance of density for a second layer of water molecules that do not directly contact the drug, but rather interact with the stabilized water molecules in the first layer. This is most clearly seen for tetracycline and kasugamycin, where multiple waters generate an extensive second layer (Fig. 2), but also

for apramycin, hygromycin B, streptomycin, tiamulin and lincomycin, where stabilization of individual second-layer water molecules is observed (Figs. 2 and 3).

## Structural conservation of antibiotic binding to ribosomes
The antibiotic streptomycin was included in two distinct complexes, which were collected at different cryo-EM facilities, and reconstructed and modeled independently. The resulting cryo-EM maps of streptomycin bound to the SSU body were obtained at 1.8 Å (Fig. 2) and 2.0 Å (Fig. 4a), respectively. Comparison of the two streptomycin–SSU structures revealed that, within the limits of the resolution, the binding modes, including direct and water-mediated interactions, were identical, indicating that our analyses are highly reproducible (Fig. 4b). We note that the binding mode of streptomycin on the *E. coli* 70S ribosome determined here is also highly similar (including the presence of the

hydrated gem-diol state of streptomycin) to that reported recently on the human mitochondrial SSU body at 2.23 Å (ref. 11) (Fig. 4c), thereby illustrating the conservation of binding of streptomycin to ribosomes that are evolutionarily distant. A big difference between the two structures is the interaction of ring II of Str with a second $Mg^{2+}$ ion in the mitoribosome that is not present in bacterial ribosomes, probably due to differences in uS12. Such differences represent attractive areas for future development of streptomycin derivatives that exhibit fewer side-effects due to interaction with human mitoribosomes[11].

In many instances, our analysis included more than one antibiotic member from the same family, such as for the tetracyclines, orthosomycins, lincosamides and pleuromutilins (Fig. 4d–l). For the tetracycline family, we included one pentacycline that bears an additional ring E, as well as two clinically approved third generation glycylcyclines, eravacycline and omadacycline, which were developed in response to increasing tetracycline resistance[4,34], and differ from tetracycline by having extensions on the C9 position (Fig. 1). In the structures of omadacycline and eravacycline in complex with the SSU head at 2.1 and 2.2 Å (Fig. 4d,e), an analogous interaction pattern was observed to that of tetracycline (Fig. 4f), consistent with the high similarity of the shared tetracyclic scaffold (Fig. 1). The high structural conservation also encompassed the water-mediated interactions as well as the position of second-layer water molecules (Fig. 4d–f). Similarly, the structure of the pentacycline on the SSU head at 1.9 Å (Extended Data Fig. 8a) revealed high structural conservation with tetracycline, eravacycline and omadacycline (Extended Data Fig. 8b–d), with expanded stacking interaction between rings D and E and C1054 of the 16S rRNA. However, despite conserved scaffolds, some variability is observed within the conformation around ring A, which influences how the Mg2 is coordinated. For omadacycline and pentacycline, the conformation allows direct coordination of Mg2 by the phosphate-oxygen(s) of G966 (Extended Data Fig. 8e,f), whereas for tetracycline and eravacycline, the Mg2–G966 coordination occurs indirectly via water molecules (Fig. 2 and Fig. 4d). A structural basis for these differences is not clear, suggesting that some conformational flexibility in this region is tolerated without affecting biological activity.

For the orthosomycins, we included evernimicin in our analysis, which contains a nearly identical heptasaccharide core to avilamycin, but is branched at ring B and contains an additional terminal benzyl moiety (ring I) (Fig. 1)[38]. The structure of evernimicin bound to the LSU at 2.0 Å reveals extensive interactions of ring D-E with the minor groove of H89 and rings F and G with H91 (Extended Data Fig. 8g), which is consistent with the large number of resistance mutations that map to this region[39–42]. The overall binding mode of the heptasaccharide core of evernimicin is analogous to that observed for avilamycin (Extended Data Fig. 8h). A completely different conformation was observed for ring F than in previous structures[43,44] (Extended Data Fig. 8i), which does not support the direct interactions between ring F and the ribosome reported previously[44]. Curiously, the ring oxygen and two methoxy groups in ring F of evernimicin and avilamycin coordinate a water molecule on the solvent side of the drugs, which cannot contribute directly to ribosome binding, but may instead stabilize the specific conformation of the ring (Fig. 3 and Extended Data Fig. 8g).

For the lincosamides, we visualized the second-generation clindamycin, which is currently used to treat a number of bacterial infections, including methicillin-resistant *Staphylococcus aureus* (MRSA)[45]. The structure of clindamycin on the LSU at 2.0 Å reveals six hydrogen bonds directly with the 23S rRNA, five from the hydroxyls of the galactopyranoside sugar and one from the amide linker, whereas no polar interactions are observed from the propyl-pyrrolidinyl tail (Fig. 4g). The overall binding mode and direct interactions, as well as many water-mediated interactions are conserved between clindamycin and lincomycin (Fig. 4h). The major difference is the presence of three waters coordinated by the 7-OH group of lincomycin (Fig. 3), which are not observed for clindamycin because a chlorine replaces the 7-OH

group with a chirality inversion (Figs. 1 and 4g). The interaction pattern of lincomycin bound to the Gram-negative *E. coli* 70S ribosome determined here is highly similar to the recent structure of iboxamycin on *T. thermophilus* at 2.5 Å (ref. 46) (Extended Data Fig. 8j–l), but also with lincomycin bound to the Gram-positive *Listeria monocytogenes* 70S ribosome at 2.1 Å (ref. 47) where the water-mediated interactions are conserved (Fig. 4i).

For the pleuromutilins, we compared tiamulin with retapamulin, the first pleuromutilin approved for human use, displaying potent activity against Gram-positive bacteria, such as MRSA[48]. The structure of retapamulin on the LSU at 1.9 Å (Fig. 4j) reveals a similar binding mode to tiamulin (Fig. 4k), including the four hydrogen bonds with the 23S rRNA, namely, two from the C21 keto group with G2061 and two from the C11 hydroxyl with the backbone of G2505 (Figs. 3 and 4j). The two water networks are also conserved between tiamulin and retapamulin (Fig. 4k), but are distinct from those observed for tiamulin in complex with the archaeal LSU at 3.2 Å (ref. 49). We did not observe a strong interaction between the sulfur atom and G2061 that was previously reported for tiamulin and retapamulin[50,51]. Moreover, while the binding of tiamulin and retapamulin to the peptidyl-transferase center (PTC) leads to shifts in U2585 and U2506 that close the drug binding pocket[51–53], changes are not observed in other nucleotides that were proposed to contribute to this induced fit mechanism[50,51] (Fig. 4l).

## MD simulations of antibiotic–ribosome interaction

The rapid cooling to cryogenic temperatures during cryo-EM sample preparation affects the structural ensemble of macromolecules[54], which raises the question to what extent the conformations of antibiotics and specifically positions of waters identified by cryo-EM are relevant at physiological temperatures. To address this question, we carried out all-atom explicit-solvent molecular dynamics (MD) simulations of the LSU with bound lincomycin at different temperatures starting from the cryo-EM structure up to physiological temperatures. The lincomycin bound structure was chosen because it has the highest resolution and contains five well-defined water molecules within hydrogen bonding distance of the antibiotic (Fig. 3). For each temperature, ten simulations were started, resulting in a total simulation time of 5 µs. As expected, the fluctuation of atomic positions of the antibiotic, measured by root mean square fluctuation (r.m.s.f.), increases with temperature (Fig. 5a, upper panel). However, the average structures of lincomycin stay very close to the cryo-EM structure, which is shown by their deviation from the cryo-EM structure (root mean square deviation, r.m.s.d. < 1 Å; Fig. 5a, lower panel). Lincomycin can be divided into two parts: galactopyranoside sugar and propyl-pyrrolidinyl tail (Fig. 5b). The galactopyranoside sugar part forms direct hydrogen bonds with rRNA nucleotides while the propyl-pyrrolidinyl tail part only has van der Waals interactions with the rRNA (Fig. 3), suggesting that the galactopyranoside sugar part is more tightly bound. Indeed, the propyl-pyrrolidinyl tail part becomes more mobile compared to the galactopyranoside sugar part with increasing temperatures highlighting the importance of direct hydrogen bonds for binding. To investigate the effect of temperature on the water molecules, we calculated the fluctuations of water positions and their distance with respect to the cryo-EM positions (Fig. 5c). The observation that, irrespective of the temperature, these distances remain small suggests that water positions identified by cryo-EM at cryogenic temperatures are stable at higher temperatures. Generally, the water fluctuations are in the same range as the lincomycin fluctuations, which would be expected if water molecules are in stable positions relative to lincomycin. In the simulations, water molecules 1 and 2 show particularly small fluctuations. These waters interact with nucleotides A2058 and A2059 (Fig. 5b), which in turn form hydrogen bonds with the less flexible galactopyranoside sugar part of lincomycin. Waters 3 and 5 that are toward the more mobile propyl-pyrrolidinyl tail part, also fluctuate more in the simulations. Water 4 shows the largest fluctuations and appears to be less tightly bound, as reflected by

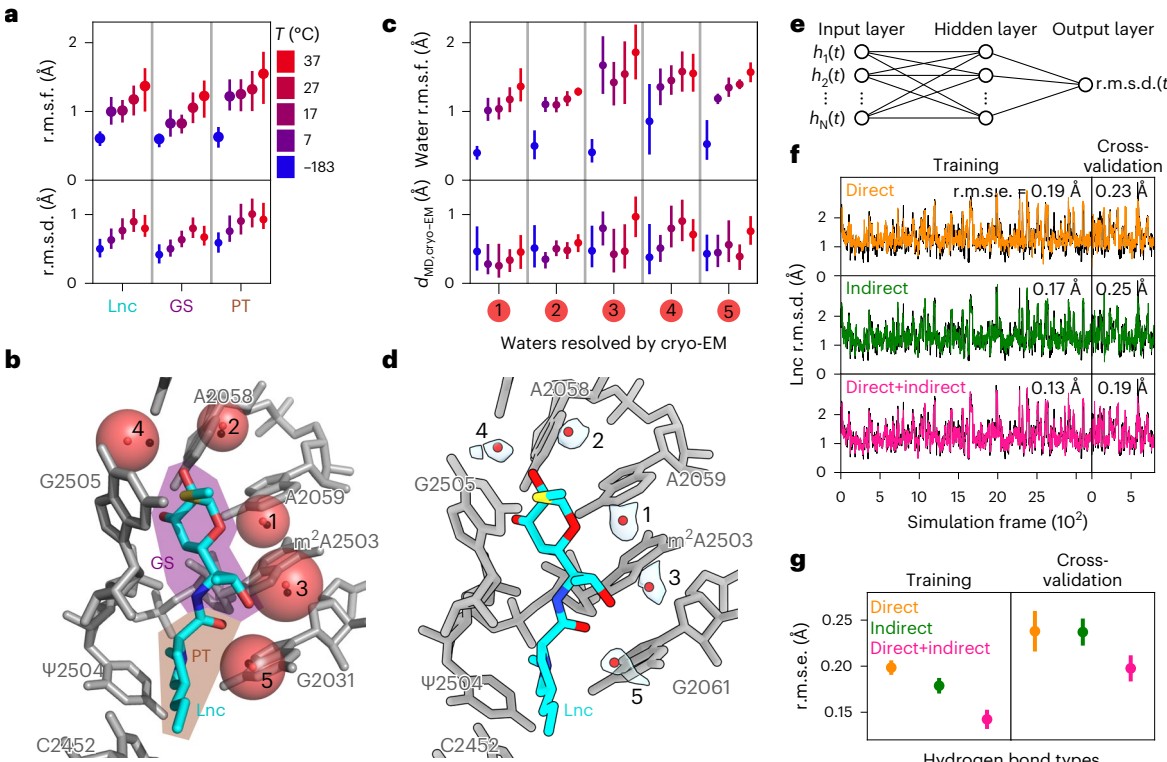

**Fig. 5 | Temperature-dependent dynamics of lincomycin and waters obtained from MD simulations. a**, Fluctuations of atom positions in simulations at different temperatures are quantified by r.m.s.f. for all lincomycin atoms (Lnc), for the galactopyranoside sugar (GS) and the propyl-pyrrolidinyl tail (PT). Structural deviations of average MD structures from cryo-EM structure are quantified by r.m.s.d. Circles and lines indicate median values and 95% confidence intervals obtained from ten independent simulations. **b**, Molecular model of lincomycin, surrounding rRNA nucleotides and waters (black spheres) within hydrogen bond distance resolved by cryo-EM. Average water positions (solid red spheres) and their r.m.s.f. (radius of transparent red spheres) obtained from MD simulations at 37 °C are shown. **c**, The r.m.s.f. for each water shown

in **b** and distances between positions in cryo-EM and MD simulations. Median and confidence intervals as described in **a**. **d**, same view as **b**, but with cryo-EM density (transparent) shown for water molecules. **e**, Neural network to predict r.m.s.d. of lincomycin from occupancies of hydrogen bonds $h_n(t)$ **f**, The r.m.s.d. of lincomycin obtained from MD simulations (black lines) and predicted by neural networks trained on occupancies of direct, indirect or direct and indirect hydrogen bonds (colors). Data used for training and cross-validation are indicated. R.m.s.e. between r.m.s.d. values predicted from the neural network and obtained from simulation. **g**, Mean (circles) and standard deviations (lines) of r.m.s.e. for training and cross-validation for ten independent neural networks with different sets of training and cross-validation data.

relatively weak density for this molecule (Fig. 5d). Taken together, these results indicate that the interplay of interactions between the antibiotic, nucleotides and water molecules contributes to antibiotic binding.

To address the question whether hydrogen bonds mediated by water molecules affect the conformation of the antibiotic, we used a multilayer perceptron neural network (Fig. 5e). The network uses hydrogen bond occupancies at simulation time points as an input and was trained to predict the structural deviation r.m.s.d. of lincomycin from the cryo-EM structure. 80% of the simulation frames were randomly chosen for training and the remaining frames were used for cross-validation (Fig. 5f). To test to what extent direct rRNA-lincomycin and indirect (water-mediated) hydrogen bonds determine the conformation, we trained the network on only direct, only indirect and on both types of hydrogen bond. To estimate the uncertainty of the predictions, the network was trained ten times on different random divisions into training and validation sets and the root mean square error (r.m.s.e.) between the predicted r.m.s.d. values and those obtained from the simulations was calculated (Fig. 5g). The predictions based on either direct or indirect hydrogen bonds have a similar accuracy showing that both hydrogen bond types contain information on the conformation of the antibiotic. The observation that networks trained on both hydrogen bond types perform better than those trained on the individual types shows that the information contained in the two hydrogen bond types is not redundant. This result indicates that the waters that mediate

hydrogen bond interactions between the antibiotic and rRNA nucleotides indeed affect the conformation of the antibiotic in its binding site.

## Discussion

Here we have determined structures of 17 distinct compounds from six different antibiotic classes bound to the *E. coli* 70S ribosome at resolutions ranging from 1.6 to 2.2 Å. This encompasses clinically relevant antibiotic families, including the tetracyclines, aminoglycosides, tuberactinomycins, pleuromutilins and lincosamides. The high quality of the structures enables a precise description of direct hydrogen bond interactions between the drug and the ribosome, but also indirect interactions that are mediated by resolved water molecules and ions (Figs. 2–4, Supplementary Videos 1–11 and Supplementary Figs. 1–17). This latter point is exemplified by tetracycline, where we observe six-atom coordination of the primary magnesium ion (Mg1) with octahedral geometry, which was less defined in previous structures (Extended Data Fig. 6), but also by the binding of hygromycin B to the SSU, where eight atoms are involved in the coordination a putative potassium ion (Fig. 2), which was not reported previously. Our study also confirmed that antibiotics with related scaffolds, that is compounds from the same antibiotic class, use identical or near-identical binding modes to interact with the ribosome (Fig. 4 and Extended Data Fig. 8). This finding contrasts with the variability that is observed when comparing previous ribosome structures of the same, or related,

antibiotics (Extended Data Fig. 1). Even slight differences in drug position and/or conformation, combined with shifts of nucleotides that comprise the binding site, can result in completely different interaction networks. The variability observed in the previous structures most likely stems from the lower precision in the models that could be generated at the given resolution. However, in some cases, we cannot exclude that observed differences arise because the structure of the antibiotic was determined on ribosomes from different bacterial species. For the antibiotics analyzed in this study, the ribosomal binding sites are highly conserved, suggesting that the interactions observed here on *E. coli* ribosomes are likely to be conserved on ribosomes from other organisms. This is also supported by the high similarity between the structures determined here of streptomycin, lincomycin and spectinomycin on the *E. coli* ribosome with recent sub-2.5 Å structures on the human mitochondrial (Fig. 4c)[11], *L. monocytogenes* (Fig. 4i)[47] and *E. faecalis* ribosomes (Extended Data Fig. 9a–c)[8], respectively. A systematic structural analysis of antibiotic–ribosome complexes from diverse species will be required to investigate this hypothesis further.

Another factor that could contribute to differences between antibiotic–ribosome structures is the functional state of the ribosome. It is conceivable that conformational changes within specific ribosomal functional states, as well as the presence of additional ligands, such as messenger RNA, tRNA or protein factors could alter the binding modes of drugs. Although individual structures of antibiotic-stalled ribosome complexes would be required to comprehensively address this point, we note a striking similarity between the binding mode of spectinomycin and apramycin determined here on vacant *E. coli* ribosomes with the same antibiotics visualized within *E. coli* translocation complexes at 2.54 (ref. 55) and 2.35 Å (ref. 7), respectively (Extended Data Fig. 9d–i). Similarly, the same direct and water-mediated interactions are observed for the common moieties of gentamicin determined here on a vacant *E. coli* 70S ribosome as for paromomycin on an *E. coli* 70S ribosome bearing mRNA, A- and P-site transfer RNAs[12] (Extended Data Fig. 9j–l). Therefore, we believe that the binding mode of antibiotics observed here on vacant *E. coli* ribosomes is conserved in most, if not all, functional complexes, and likely represents the initial binding mode of the drug to the ribosome before translational arrest. Additional ligands and/or distinct conformations may lead to local changes in binding patterns but are not likely to affect the core interactions described in this work.

Because the cryo-EM structures are determined at cryogenic temperatures (−180 °C), we also investigated whether the observed water molecules would be ordered at higher, more physiological, temperatures (37 °C). MD simulations using the best resolved (1.64 Å) lincomycin–LSU structure were performed with a range of different temperatures (−180 to 37 °C), revealing that all water molecules remained stably bound at 37 °C, but that increased fluctuations were observed for both the drug and water molecules at increased temperatures (Fig. 4a–c). The extent of the fluctuations varied for the different water molecules (Fig. 4c), and correlated well with the intensity of the cryo-EM map density (Fig. 4d), which suggests a greater contribution of waters W1 and W2 for drug binding compared with W3–W5. Indeed, our neural network analysis suggests that both direct as well as indirect water-mediated interactions between the drug and the ribosome contribute to the conformation of the antibiotic in its binding site.

A major finding of our study is that the antibiotics are highly solvated within their ribosomal binding sites, and that the level of solvation varies dramatically for the different antibiotic classes (Figs. 2 and 3). We observe that many antibiotics displace bound waters on ribosome binding, but also form additional interactions via coordination of preordered waters, as illustrated for tetracyclines, streptomycins, spectinomycin and kasugamycin (Extended Data Fig. 7a–d). This contrasts with the drug binding sites at the PTC on the LSU, which are

relatively free of ordered waters. In these cases, all water-mediated interactions observed for PTC-targeting antibiotics, such as the pleuromutilins or lincosamides analyzed here, arise because of stabilization of water molecules on drug binding (Extended Data Fig. 7e–h). Thus, the improved resolution of the presented structures allows us to distinguish between entropic and enthalpic contributions to drug binding.

In conclusion, we present a high-precision atlas of antibiotic interaction with ribosomes that encompasses the visualization of water- and ion-mediated networks. We envisage that such information can be used in the future for structure-based design of new antibiotic derivatives by identifying regions within the compounds that can be altered to displace stably bound waters and assume their interactions with the target. Water displacement from a binding site has been shown to have a favorable effect on drug affinity, due to the entropic gain when the surface-associated solvent molecules are released into the bulk solvent[15,18]. Additionally, it is conceivable that compound modifications can be designed to establish additional interactions with preordered water molecules to gain additional binding energy outside the original binding site[15,18]. Moreover, our study provides fundamental insight into small molecule interaction with RNA, which is likely to be important for the development of other therapeutically relevant RNA-targeting ligands[22–24].

## Online content

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

## Methods

### Preparation of antibiotic–ribosome complexes

In vitro reconstituted *E. coli* 70S ribosomes were generated from the *E. coli* K12 strain BW25113, as described previously[56]. Antibiotic–ribosome samples were generated by incubating antibiotic cocktails 1–5 with *E. coli* 70S ribosomes in buffer A (50 mM HEPES-KOH, pH 7.5, 25 mM Mg(OAc)$_2$, 80 mM NH$_4$Cl, 100 mM KOAc, 1 mM DTT, 0.05% DDM) at 37 °C for 15 min, before being frozen at −80 °C until use. Final antibiotic concentrations for complexes formed with each cocktail was: cocktail 1 contained 200 μM omadacycline (MedChemExpress), 200 μM spectinomycin (Santa Cruz Biotechnology), 200 μM streptomycin (Santa Cruz Biotechnology), 200 μM evernimicin, 200 μM hygromycin B (Cayman Chemical); cocktail 2 contained 100 μM capreomycin (Sigma Aldrich), 100 μM kasugamycin (Sigma Aldrich) and 100 μM retapamulin (Sigma Aldrich); cocktail 3 contained 100 μM tetracycline (Sigma Aldrich), 100 μM viomycin (Sigma Aldrich), 100 μM streptomycin (Santa Cruz Biotechnology), 100 μM lincomyin (Sigma Aldrich) and 100 μM avilamycin (Cayman Chemical); cocktail 4 contained 10 μM apramycin (Sigma Aldrich), 10 μM eravacycline (MedChemExpress) and 100 μM clindamycin (Santa Cruz Biotechnology); cocktail 5 contained 100 μM pentacycline (Tetraphase), 10 μM gentamicin (Carl Roth) and 100 μM tiamulin (Sigma Aldrich).

### Preparation of cryo-EM grids and data collection

Here, 3.5 μl (7 optical density (OD$_{260nm}$)/ml) of each antibiotic–ribosome complex 1–5 were applied to freshly plasma-cleaned graphene coated TEM grids (Quantifoil, Au, 300 mesh, R1.2/1.3). Graphene coating was carried out according to the in-house optimized protocol for transfer of monolayer chemical vapor deposition graphene to the grid surface. The graphene grids were then hydrophilized in H:O plasma (40:1) for 30 s using Gatan Solarus II immediately before use. Sample vitrification into liquid ethane was performed using a Thermo Scientific Vitrobot Mark IV (4 °C, 100% rel. humidity, 30 s waiting time, 3 s blotting time). The grids were subsequently mounted into the Autogrid cartridges and loaded to Talos Arctica (Thermo Scientific) TEM for screening. Grids were stored in liquid nitrogen until high-resolution data collection.

### Data acquisition

**Dataset 1.** Single particle cryo-EM data were collected in automated manner on Titan Krios G1 (FEI-Thermo Scientific) TEM operated at 300 kV using SerialEM software[57]. An example micrograph is shown in Supplementary Fig. 18a. The microscope was aligned for fringe-free imaging and equipped with K2 (Ametek) direct electron detector. The camera was operated in electron counting mode and the data were collected at the pixel size of 0.53 Å px$^{-1}$. The microscope condenser system was set to produce 11 e$^-$/Å$^2$ s electron flux on the specimen and the data from 4.0 s exposure were stored into 40 frames. The energy selecting slit was set to 10 eV. The data from 3 × 3 neighboring holes were collected using beam and/or image shifting while compensating for the additional coma aberration. The data were collected with the nominal defocus range of −0.4 to −1.6 μm. Total number of 24,195 videos was collected within a 96-hour session.

**Datasets 2–4.** Single particle cryo-EM data were collected in automated manner using EPU v.3.0 on a cold-FEG fringe-free Titan Krios G4 (FEI-Thermo Scientific) TEM operating at 300 kV equipped with a SelectrisX energy filter and a Falcon IV. Example micrographs are shown in Supplementary Fig. 18b–d. The camera was operated in electron counting mode with the energy selecting slit set to 10 eV and the data were collected at the pixel size of 0.45 Å px$^{-1}$. The microscope condenser system was set to produce 6 e$^-$ px$^{-1}$ s$^{-1}$ electron flux on the specimen and the data from 4.5 s exposure were stored as .EER files. The data from 3 × 3 neighboring holes were collected using beam and/or image shifting while compensating for the additional coma aberration. The data were collected with the nominal defocus range of −0.4 to −1 μm.

With an average of 500 images per hour, a total number of 21,971 videos for dataset 2, 33,815 videos for dataset 3 and 38,434 videos for dataset 4 were collected within one 48-h and two 72-h sessions, respectively.

**Dataset 5.** The data were collected on the same Titan Krios microscope as for dataset 1. The microscope was aligned for fringe-free imaging and equipped with Bioquantum K3 (Ametek) direct electron detector. The camera was operated in correlated double sampling mode and the data were collected at the pixel size of 0.51 Å px$^{-1}$. The microscope condenser system was set to produce 25 e$^-$/Å$^2$ s electron flux on the specimen and the data from 1.8 s exposure were stored into 40 frames. The energy selecting slit was set to 10 eV. The data from 3 × 3 neighboring holes were collected using beam and/or image shifting while compensating for the additional coma aberration. The data were collected with the nominal defocus range of −0.4 to −1.6 μm. Total number of 37,094 videos was collected within a 72-h session. An example micrograph is shown in Supplementary Fig. 18e.

### Cryo-EM data processing

Processing was performed in RELION v.3.1 (dataset 1 only) or v.4 (ref. 25). MotionCor2 (ref. 58) with 5 × 5 patches and CTFFIND4 (ref. 59) (using power spectra) were used for motion correction and initial contrast transfer function (CTF) estimation, unless otherwise specified. The resolution of CTF fits and CTF figure of merit were used to remove outlier micrographs. Particles were picked with crYOLO[60], or Topaz[61] within RELION. After 2D classification, all ribosome-like classes were selected, particles were extracted with a pixel size of roughly 2.5 Å and a volume was reconstructed ab initio. After 3D refinement using the ab initio volume as a reference[62], 3D classification without angular sampling was performed. All classes that contained 70S ribosomes at high resolution were used for further processing. Particles were re-extracted with a smaller pixel size, subjected to 3D auto-refinement and CTF refinements were performed to correct for anisotropic magnification, trefoil and higher-order aberration corrections, defocus and astigmatism, followed by 3D auto-refinement and Bayesian polishing[63] and another 3D auto-refinement. CTF and 3D auto-refinements were repeated until no further improvement in resolution was obtained. Masks around the regions of interest were created and used for partial signal subtraction. For volumes with resolutions beyond roughly 1.8 Å, the volumes are reconstructed with Ewald sphere correction. For multibody or focused refinements, volumes corresponding to the LSU core, SSU body and SSU head were isolated using the volume eraser tool in UCSF ChimeraX[64], and masks created from the densities low-pass filtered to 30 Å. RELION 4 (ref. 25) was used to estimate local resolution (Extended Data Figs. 2–4).

For dataset 1, micrographs were grouped into nine optics groups according to image shift position. Micrographs with an estimated MaxRes more than 5 Å, a CTF figure of merit less than 0.1 and/or crystalline ice rings visible in power spectra were discarded, resulting in 23,361 micrographs. crYOLO[60] was used for picking, resulting in 754,663 particles. For 2D classification, CTFs were ignored until the first peaks and the maxsig parameter set to 5. All classes containing ribosomal particles were selected. A subset of classes that contained apparent noisy particles were selected for another round of 2D classification with the same settings as above. Combined, this resulted in 536,799 particles. After 3D classification, 514,855 particles were selected from for further processing. Particles were re-extracted with a pixel size of 0.762 for further processing.

For dataset 2, MotionCor2 (ref. 58) with 4 × 4 patches was initially used to align micrograph videos. Micrographs with an estimated MaxRes more than 15 Å were discarded, resulting in 17,197 micrographs. crYOLO[60] was used for picking, resulting in 219,953 particles. For 2D classification all classes containing ribosomal particles were selected, this resulted in 179,724 particles. Particles were re-extracted with a pixel size of 0.72 Å and a box size of 720$^2$ pixels for further processing and

subject to 3D auto-refinement, CTF refinements, and Bayesian polishing and partial signal subtraction as described above.

For dataset 3, MotionCor2 (ref. [58]) with 4 × 4 patches was initially used to align micrograph videos. After CTF estimation, cutoffs of MaxRes more than 8 Å and CTF figure of merit less than 0.05 were applied, resulting in 28,487 micrographs used for subsequent processing. Then, 757,044 particles were picked with Topaz[61]. For 2D classification, CTFs were ignored until the first peaks, and the maxsig parameter set to 5. All classes containing ribosomal particles were selected. A subset of classes that contained apparent noisy particles were selected for another round of 2D classification with the same settings as above. Combined, this resulted in 442,159 particles that were used for further processing. After 3D classification, 419,159 particles were processed further. Initially, particles were re-extracted with a pixel size of 0.768 Å and a box size of 600 × 600 pixels and subject to 3D auto-refinement, CTF refinements and Bayesian polishing as described above. Another round of Bayesian polishing was performed with particles extracted with a pixel size of 0.681 Å and a box size of 800 × 800 pixels before final 3D auto-refinements. B factors estimated by Guinier analysis were implausibly small with poor correlations of fit. B factors were estimated instead by taking random subsets of particles and plotting the resolution after 3D auto-refinement against number of particles[65].

For dataset 4, MotionCor2 (ref. [58]) with 4 × 4 patches was initially used to align 38,434 micrograph videos. After CTF estimation, cutoffs of MaxRes more than 15 Å and CTF figure of merit less than 0.05 were applied, resulting in 34,108 micrographs used for subsequent processing. Then, 464,723 particles were picked using crYOLO[60]. For 2D classification, CTFs were ignored until first peak and 410,594 ribosome-like particles were selected and processed further. Following an initial 3D auto-refinement step the particles were subjected to 200 iterations of 3D classification from which three classes containing a total of 275,137 particles were combined. The combined particles were re-extracted with a pixel size of 0.768 Å and a box size of 600 × 600 pixels and subjected to 3D auto-refinements, CTF refinements and Bayesian polishing as described above. Following a final 3D auto-refinement an automatically estimated B factor was applied during postprocessing. Focus refinements were performed individually for LSU, SSU head and body with masked particle subtraction at a pixel size of 0.768 Å.

For dataset 5, MotionCor2 (ref. [58]) with 3 × 3 patches was initially used to align micrograph videos. Two optics groups, corresponding to two different collections, were used for this dataset. After CTF estimation, cutoffs of MaxRes more than 4.5 Å and CTF figure of merit less than 0.2 were applied, resulting in 35,819 micrographs used for subsequent processing. Topaz[61] was used for picking, resulting in 2,146,827 initial particles. For 2D classification, CTFs were ignored until the first peaks and the maxsig parameter set to 5. All classes containing ribosomal particles were selected. A subset of classes that contained apparent noisy particles were selected for another round of 2D classification with the same settings as above. Combined, this resulted in 1,552,367 particles that were used for further processing. After 3D classification, 1,301,160 particles were processed further. Particles were re-extracted with a pixel size of 0.767 Å and a box of 480 × 480 pixels for further processing. For the SSU head focused refinement, particles were re-extracted with a box size of 416 × 416 pixels.

## Generation of molecular models
Initial models for structure were generated based on the available molecular model of the *E. coli* 70S ribosome at 1.98 Å (Protein Data Bank (PDB) ID 7K00)[12]. Molecular models for datasets 1, 4 and 5 were generated initially using the final refined molecular model for dataset 3. The initial models containing ribosomal proteins and rRNA were rigid body fitted to the relevant cryo-EM map density using ChimeraX[64] and then manually adjusted in Coot[66,67]. Servalcat[68] was used for model-refinement and to help identify problematic regions as well as new (or any secondary) antibiotic binding sites. Magnesium

ions were placed into visible magnesium-water clusters, potassium ions were designated according to previous models (PDB 6QNQ)[69] and in some cases on the basis of density alone, namely coordination pattern, distance to interacting atoms and strength of density compared to nearby magnesium ions. Waters were added in a semi-automated fashion using the Coot 'find waters' dialog on the difference map (Servalcat)[68]. The resulting waters were manually augmented where necessary in Coot and additional waters were placed in remaining empty densities apparent from the difference map[68]. For compounds without available 3D structures, models were generated using ChemDraw (PerkinElmer Informatics) with structural restraints generated through PRODRG2 (ref. [70]), aceDRG[71] or Phenix eLBOW[72]. The output model was manually modeled into the corresponding density with Coot[66,67]. Models were refined with metal and structural restraints calculated by Phenix eLBOW[72] or using Servalcat[68] and validated by Phenix and the comprehensive cryo-EM validation and MolProbity server[73] with map versus model cross-correlation at a Fourier shell correlation (FSC)$_{0.5}$ for all individual maps.

## Structure alignment
Structures for model comparisons were aligned using ChimeraX[64]. Initially, models were aligned globally on rRNA present in both models, then a portion of both models was selected in a 10 Å radius around the compound of interest and the alignment was repeated using the 'matchmaker' tool of ChimeraX focusing on the selected regions.

## MD simulations
To obtain the dynamics of lincomycin bound to the LSU and surrounding water molecules at different temperatures, we carried out all-atom explicit-solvent MD simulations of the complete LSU. For the simulations, the cryo-EM structure of the LSU with bound lincomycin, waters and ions were placed in a dodecahedron box with a minimum distance of 15 Å between the atoms and the box boundaries. The box was solvated using the program solvate[74]. Histidine protonation states were determined using WHAT IF[75]. The simulation system charge was neutralized by adding $K^+$ ions using GENION[74]. Next, $Mg^{2+}Cl^-$ and $K^+Cl^-$ ions were added with 7 and 150 mM concentrations, respectively. The simulations were performed using GROMACS-2022.4 (ref. [74]) with the amber14SB force field[76] and the OPC water model[77]. Parameters from Joung and Cheatham[78], Grotz et al.[79] and Aduri et al.[80] were used for $K^+Cl^-$ ions, $Mg^{2+}$ ions and modified nucleotides, respectively. The initial set of lincomycin coordinates from the cryo-EM model was protonated and energy-minimized by a HF/6-31G* optimization in GAUSSIAN 09 (Gaussian Inc., https://www.gaussian.com) and parameterized with the General Amber Force Field[81]. For General Amber Force Field parameter assignment, ACPYPE[82] and AnteChamber[83] tools were used. The particle mesh Ewald method with grid spacing of 0.12 nm and cutoff of 1 nm was used to calculate long-range electrostatics[84]. The van der Waals interactions were calculated within a 1 nm cutoff.

The simulation system was energy-minimized with harmonic position restraints, with a force constant $k = 1,000$ kJ mol$^{-1}$ nm$^{-2}$, applied to all lincomycin and ribosome atoms that were resolved in the cryo-EM map. Resolved water oxygens and ions were restrained with $k = 5,000$ kJ mol$^{-1}$ nm$^{-2}$. For each temperature (90, 280.15, 290.15, 300.15 and 310.15 K), ten independent simulations were run in three steps. In the first step, to equilibrate the added solvent, 10 ns MD simulations were carried out with the position restraints. Subsequently, the force constants $k$ were linearly decreased during 10 ns. In the third step, the simulations were continued for 100 ns without position restraints. A Berendsen barostat[85] was used in steps 1 and 2, a Parrinello–Rahman barostat[86] for step 3 (both $\tau_P = 1$ ps). In all steps, bond lengths were constrained using the LINCS algorithm[87] and an integration step of 2 fs was used. Solute and solvent temperatures were controlled independently using velocity rescaling ($\tau_T = 0.1$ ps)[88]. Atom positions were recorded every 10 ps and only the trajectories of step 3 were used for analysis.

## Analysis lincomycin and water dynamics

For all following analyses, the trajectories were aligned to the cryo-EM structure using the coordinates of atoms belonging to rRNA nucleotides within 15 Å of lincomycin. For each frame of each trajectory and for each of the five resolved water oxygens within H bond distance of lincomycin, the position of water oxygen closest to the cryo-EM position was assigned to that water oxygen. To quantify the dynamics of lincomycin and water oxygens, the r.m.s.f. was calculated. To that aim, for each temperature, the ten corresponding trajectories were concatenated and the r.m.s.f.s were calculated. The uncertainty was estimated by bootstrapping the trajectories (1,000 iterations) before concatenating, calculating r.m.s.f.s and subsequently obtaining 95% confidence intervals. For each temperature, the structural deviation of lincomycin from the cryo-EM structure was measured using the r.m.s.d. of the average structure calculated from all corresponding trajectories. Analogously, the distance of mean water positions from their position in the cryo-EM structure was calculated. The r.m.s.d. and distance uncertainties were obtained by bootstrapping trajectories (1,000 iterations).

## Neural networks to predict structural deviation from hydrogen bonds

To test whether H bonds between rRNA and lincomycin mediated by water molecules affect the conformation of lincomycin, we trained artificial neural networks to predict lincomycin r.m.s.d.s from H bond occupancies. From the trajectories, we extracted three types of hydrogen bond using the program gmx hbond[74]: between lincomycin and rRNA nucleotides, between lincomycin and waters, and between waters and rRNA nucleotides. The latter two sets were combined to hydrogen bonds between lincomycin and nucleotides mediated by one water molecule. Next, for intervals of 1 ns, hydrogen bond occupancies and r.m.s.d. of lincomycin relative to the cryo-EM structure were averaged. To avoid overfitting of neural networks, we divided the data into training and cross-validation sets. To that aim, we first divided the averaged H bond occupancies $h_n(t)$, where $n$ is the $n$th H bond and the averaged deviations r.m.s.d.$(t)$ into chunks of 10 ns. Then 80% of these chunks were randomly sorted into the training set and the remaining 20% into the cross-validation set. To estimate the uncertainties of the cross-validation, the sorting was repeated ten times and ten independent neural networks were trained on the training sets. We used multilayer perceptron neural networks with a rectified linear unit activation function and tested different numbers of neurons per layer (10, 20, 30, 40) and either one or two hidden layers. The models were implemented, trained and analyzed using the deep learning API Keras (https://keras.io), which runs on top of TensorFlow (https://tensorflow.org). To optimize the neural networks, stochastic gradient descent was used and to avoid overfitting on a specific training set, we used the early stopping option with a patience of 100 epochs and the cross-validation r.m.s.e. as the metric. Mean and standard deviations of obtained r.m.s.e.s between prediction and data the cross-validation sets for each neural network are shown in Extended Data Fig. 10. In the main text, we only discuss the network with one hidden layer consisting of 30 neurons, which was chosen because it resulted in the lowest cross-validation r.m.s.e. values.

## Figure preparation

Figures were prepared using UCSF ChimeraX[64] and Inkscape (https://inkscape.org/).

## Reporting summary

Further information on research design is available in the Nature Portfolio Reporting Summary linked to this article.

## Data availability

Initial models for structure were generated based on the *E. coli* 70S ribosome PDB ID 7K00, and potassium ions were designated according to PDB 6QNQ. The cryo-EM maps for the antibiotic–ribosome complexes have been deposited in the EM Data Bank with the accession code EMD-16520 (dataset 1, SSU head), EMD-16526 (dataset 1, SSU body), EMD-16530 (dataset 1, LSU), EMD-16536 (dataset 2, SSU head), EMD-16612 (dataset 2, SSU body), EMD-16613 (dataset 2, LSU), EMD-16615 (dataset 3, SSU head), EMD-16645 (dataset 3, SSU body), EMD-16646 (dataset 3, LSU), EMD-16620 (dataset 4, SSU head), EMD-16650 (dataset 4, SSU body), EMD-16641 (dataset 4, LSU), EMD-16644 (dataset 5, SSU head), EMD-16651 (dataset 5, SSU body) and EMD-16652 (dataset 5, LSU). The respective coordinates for the electron-microscopy-based model of the antibiotic–ribosome complexes are deposited in the PDB with the accession code PDB 8CA7 (dataset 1, SSU head), PDB 8CAI (dataset 1, SSU body), PDB 8CAM (dataset 1, LSU), PDB 8CAZ (dataset 2, SSU head), PDB 8CEP (dataset 2, SSU body), PDB 8CEU (dataset 2, LSU), PDB 8CF1 (dataset 3, SSU head), PDB 8CGJ (dataset 3, SSU body), PDB 8CGK (dataset 3, LSU), PDB 8CF8 (dataset 4, SSU head), PDB 8CGR (dataset 4, SSU body), PDB 8CGD (dataset 4, LSU), PDB 8CGI (dataset 5, SSU head), PDB 8CGU (dataset 5, SSU body) and PDB 8CGV (dataset 5, LSU).

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

## Acknowledgements

We thank K. Yamashita for his tireless improvements to Servalcat and his help resolving refinement issues in general. We acknowledge cryo-EM and tomography core facility CEITEC MU of CIISB, Instruct-CZ Centre supported by MEYS CR (grant no. LM2023042), iNEXT-Discovery, project number 871037 and the European Regional Development Fund-Project "UP CIIB" (No. CZ.02.1.01/0.0/0.0/18_046/0015974). This research was funded by the Deutsche Zentrum für Luft-und Raumfahrt (grant no. DLR01Kl1820 to D.N.W.) within the RIBOTARGET consortium under the framework of JPIAMR and by the Deutsche Forschungsgemeinschaft (the German Research Foundation) grant no. WI3285/12-1 (to D.N.W.) and under Germany's Excellence Strategy grant no. EXC 2067/1-390729940 (L.V.B and H.G.). The funders had no role in study design, data collection and analysis, decision to publish or preparation of the manuscript.

## Author contributions

D.N.W. designed the study. M.M. prepared the cryo-EM samples. J.N., B.B. and A.G.M. collected the cryo-EM data. H.P., C.C.-M. and T.O.K. processed the cryo-EM data. H.P. and C.C.-M. built and refined the molecular models. L.V.B. and H.G. conducted the MD simulations. D.N.W. wrote the paper with help from all authors.

## Funding

## Competing interests

The authors declare no competing interests.

## Additional information

**Extended data** is available for this paper at https://doi.org/10.1038/s41594-023-01047-y.

**Correspondence and requests for materials** should be addressed to Daniel N. Wilson.

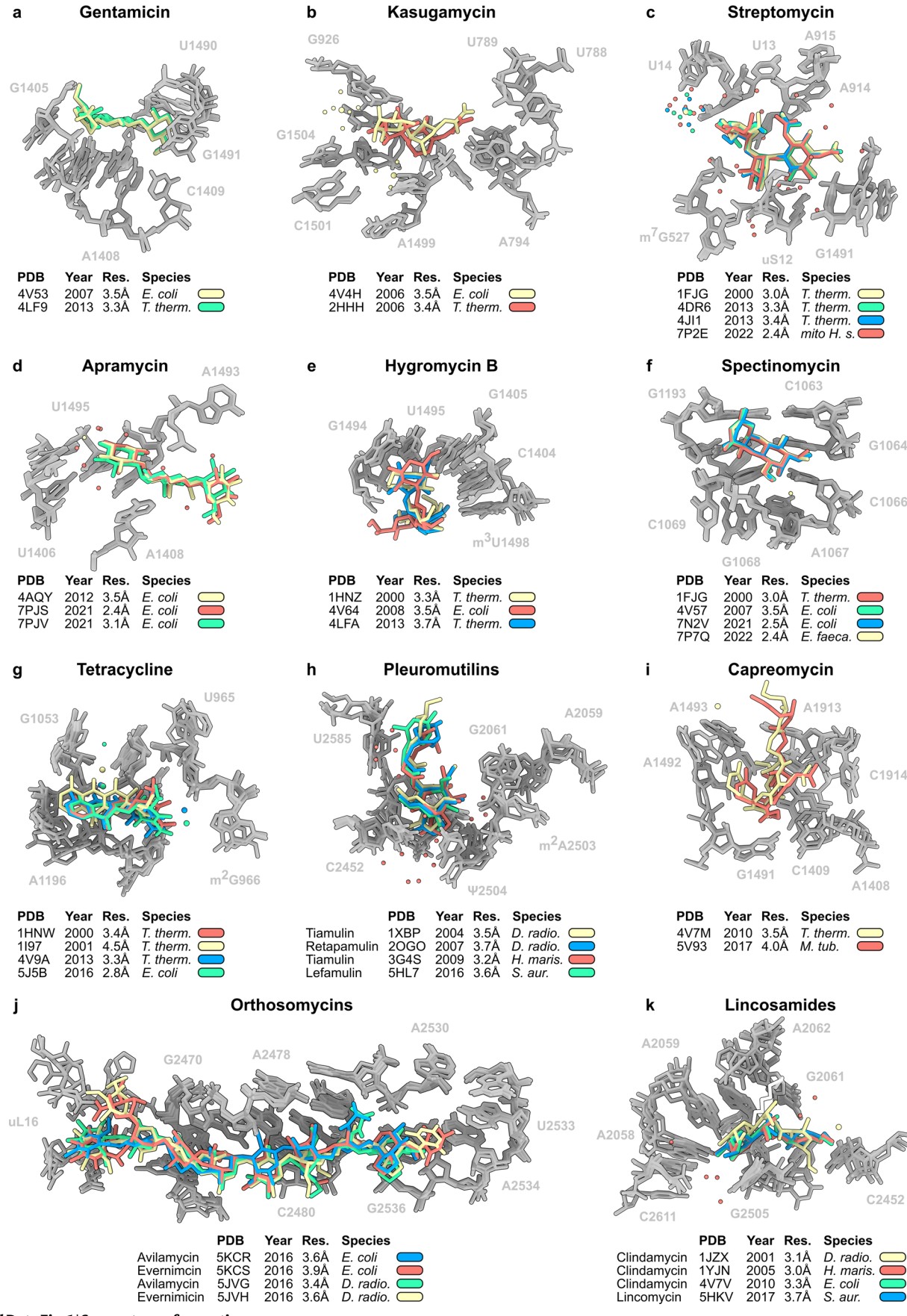

**Extended Data Fig. 1 | See next page for caption.**

**Extended Data Fig. 1 | Comparison of structures of antibiotic-ribosome complexes. a-k**, Superimposition of previous structures of diverse antibiotic-ribosome complexes, including (**a**) gentamicin (yellow) on the *E. coli* 70S ribosome at 3.5 (PDB ID 4V53)[28] with gentamicin (green) on the *T. thermophilus* 30S at 3.3 (PDB ID 4LF9), (**b**) kasugamycin (yellow) on the *E. coli* 70S ribosome at 3.5 Å (PDB ID 4V4H)[89] with kasugamycin (red) on the *T. thermophilus* 30S at 3.4 Å (PDB ID 2HHH)[30], (**c**) streptomycin on the *T. thermophilus* 30S at 3.0 Å (yellow; PDB ID 1FJG)[90] at 3.3 Å (green, PDB ID 4DR6)[91] at 3.35 (blue, PDB ID 4JI1)[91] with streptomycin (red) on the human mitochondrial small subunit at 2.4 Å (PDB ID 7P2E)[92], (**d**) apramycin (yellow) on the *T. thermophilus* 30S at 3.5 Å (PDB ID 4AQY)[29] with apramycin on the *E. coli* 70S ribosome at 2.4 Å (red, PDB ID 7PJS)[7] and 3.1 Å (green, PDB ID 7PJV)[7], (**e**) hygromycin B on the *T. thermophilus* 30S at 3.3 Å (yellow, PDB ID 1HNZ)[26] and 3.7 Å (blue, PDB ID 4LFA) with hygromycin B on the *E. coli* 70S ribosome at 3.5 Å (red, PDB ID 4V64)[37], (**f**) spectinomycin (red) on the *T. thermophilus* 30S at 3.0 Å (PDB ID 1FJG)[90] with spectinomycin on *E. coli* 70S ribosome at 3.5 Å (green, PDB ID 4V57)[28], *E. faecalis* 70S ribosome at 2.4 Å (yellow, PDB ID 7P7Q)[8] and within an *E. coli* 70S translocation intermediate at 2.5 Å (blue, PDB ID 7N2V)[55], (**g**) tetracycline on the *T. thermophilus* 30S at 3.4 Å (red, PDB ID 1HNW)[26] and 4.5 Å (yellow, PDB ID 1I97)[27] with tetracycline (blue) on the *T. thermophilus* 70S at 3.3 (PDB ID 4V9A)[93] and tetracycline (green) on the *E. coli* 70S at 2.8 Å (PDB ID 5J5B)[94], (**h**) the pleuromutilins tiamulin (yellow) on the *D. radiodurans* 50S at 3.5 Å (PDB ID 1XBP)[50], retapamulin (blue) on the *D. radiodurans* 50S at 3.7 Å (PDB ID 2OGO)[51], tiamulin (red) on the archaeal *H. marismortui* 50S at 3.2 Å (PDB ID 3G4S)[49] and lefamulin (green) on the *S. aureus* 50S at 3.6 Å (PDB ID 5HL7)[53], (**i**) capreomycin (yellow) on the *T. thermophilus* 70S at 3.5 Å (PDB ID 4V7M)[32] with capreomycin (red) on the *M. tuberculosis* 70S at 4.0 Å (PDB ID 5V93)[95], (**j**) the orthosomycins avilamycin (blue, PDB ID 5KCR) and evernimicin (red, PDB ID 5KCS) on the *E. coli* 70S at 3.9 Å[43] with avilamycin (green, PDB ID 5JVG) and evernimicin (yellow, PDB ID 5JVH) on the *D. radiodurans* 50S at 3.6 Å and 3.4 Å, respectively[44], (**k**) the lincosamide clindamycin on the *D. radiodurans* 50S at 3.1 Å (yellow, PDB ID 1JZY)[96], on the *H. marismortui* 50S at 3.0 Å (red, PDB ID 1YJN)[97] and on the *E. coli* 70S at 3.3 Å (green, PDB ID 4V7V)[98] with lincomycin on the *S. aureus* 50S at 3.7 Å (blue, PDB ID 5HKV)[99]. Alignments were made using the rRNA within 10 Å of the antibiotic. rRNA and r-proteins comprising the binding site are colored grey, whereas antibiotics (including waters and ions if present) are color-coded as indicated.

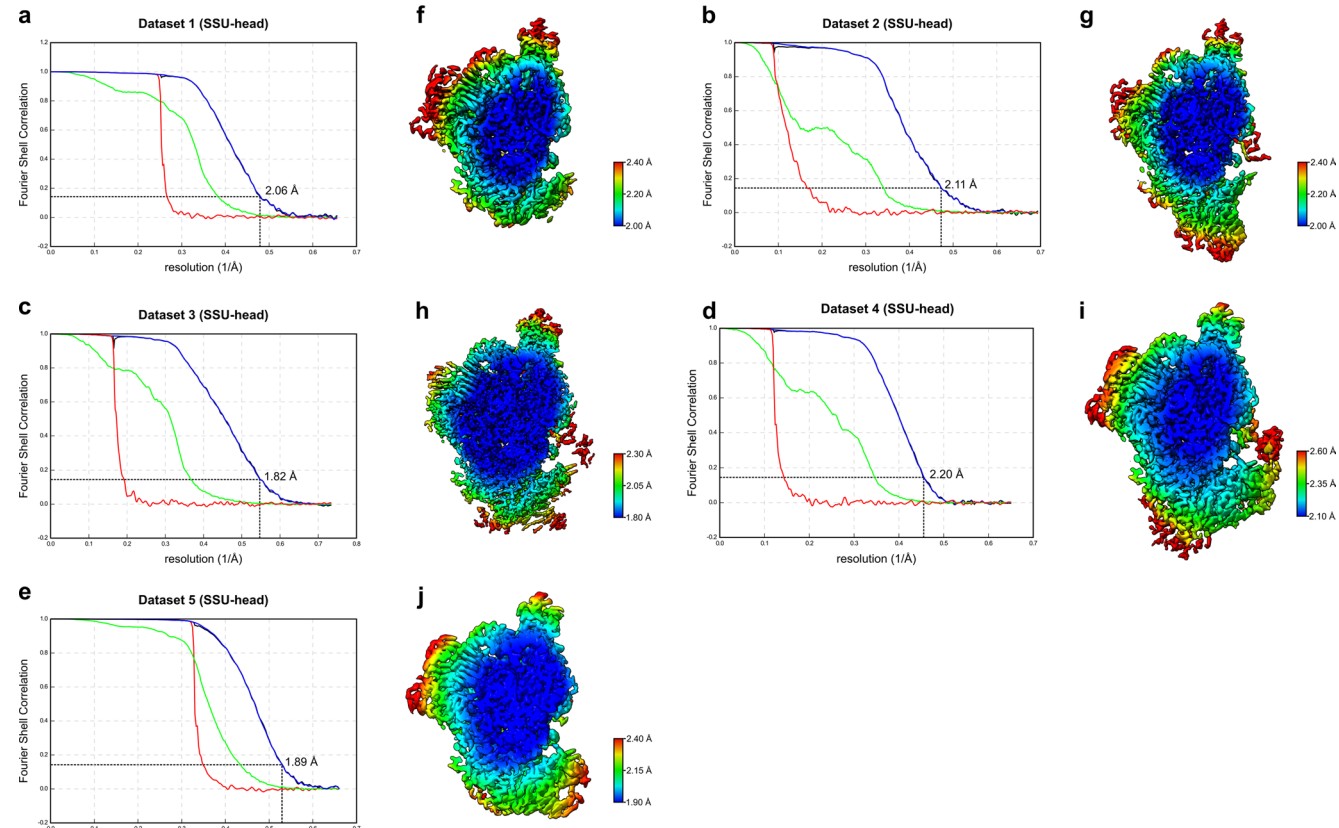

**Extended Data Fig. 2 | Resolution of antibiotic-SSU head maps. a-e**, Fourier-shell correlation (FSC) curve of the focused refined antibiotic-SSU head maps for (**a**) dataset 1, (**b**) dataset 2, (**c**) dataset 3, (**d**) dataset 4, and (**e**) dataset 5. The different curves include the masked map (green), unmasked map (blue) and the phase-randomized masked map (red). The dashed line at FSC 0.143 indicates that the average resolution for the SSU head maps for datasets 1–5 are 2.06 Å, 2.11 Å, 1.82 Å, 2.20 Å, 1.89 Å, respectively. **f-j**, Transverse section of the SSU head colored by local resolution for the focused refined maps calculated from (**f**) dataset 1, (**g**) dataset 2, (**h**) dataset 3, (**i**) dataset 4, and (**j**) dataset 5.

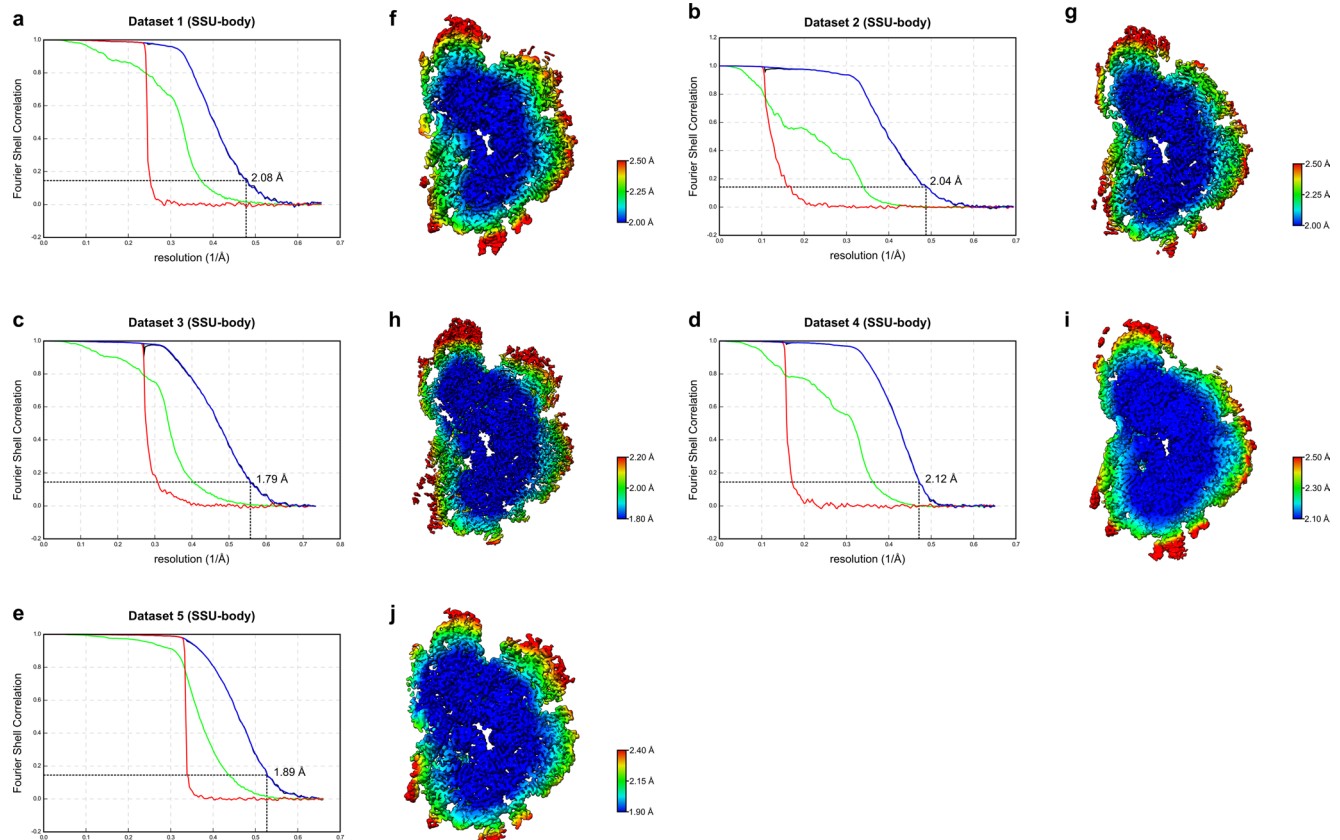

**Extended Data Fig. 3 | Resolution of antibiotic-SSU body maps.**
**a-e**, Fourier-shell correlation (FSC) curve of the focused refined SSU body maps for (**a**) dataset 1, (**b**) dataset 2, (**c**) dataset 3, (**d**) dataset 4, and (**e**) dataset 5. The different curves include the masked map (green), unmasked map (blue) and the phase⁻randomized masked map (red). The dashed line at FSC 0.143 indicates that the average resolution for the SSU body maps for datasets 1–5 are 2.08 Å, 2.04 Å, 1.79 Å, 2.12 Å, 1.89 Å, respectively. **f-j**, Transverse section of the SSU body coloured by local resolution for the focused refined maps calculated from (**f**) dataset 1, (**g**) dataset 2, (**h**) dataset 3, (**i**) dataset 4, and (**j**) dataset 5.

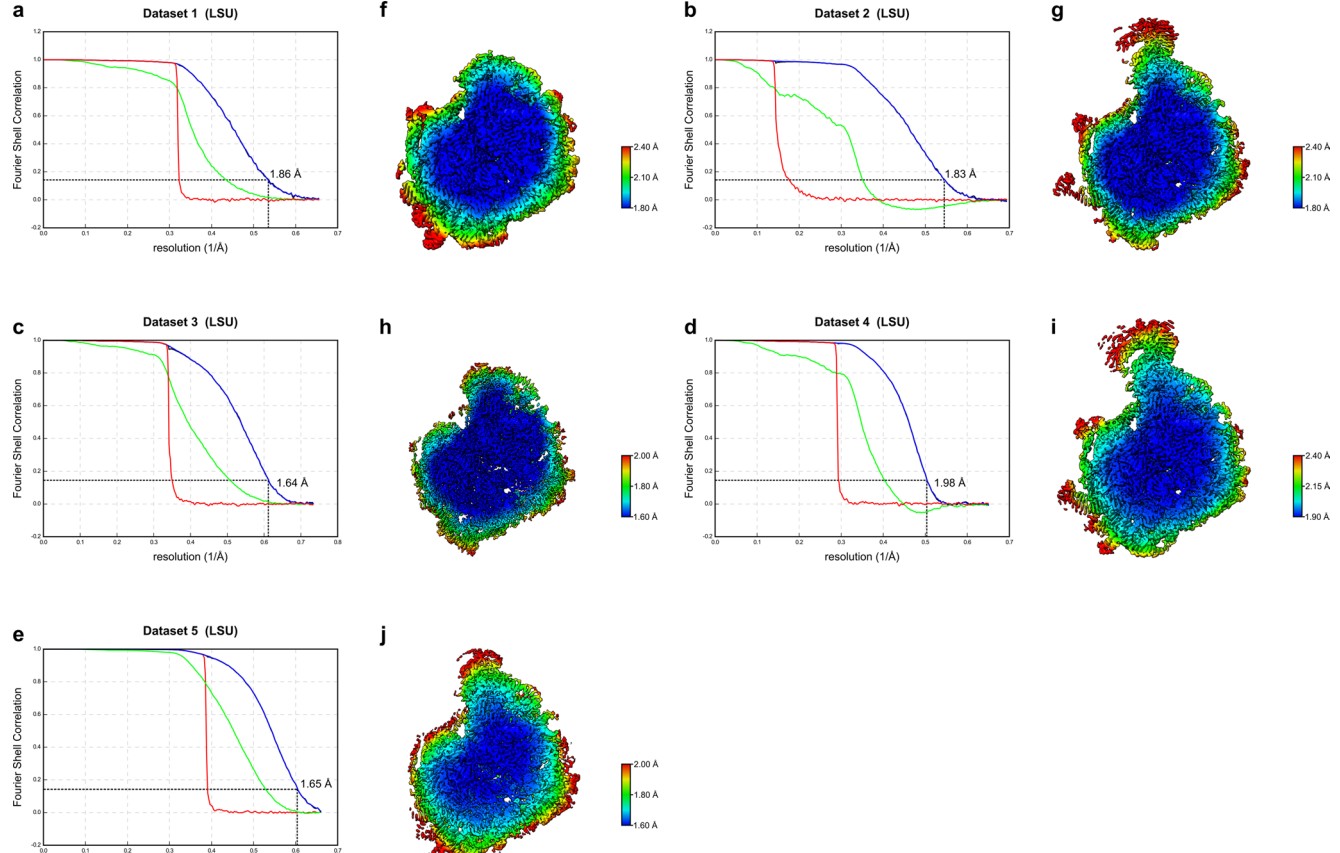

**Extended Data Fig. 4 | Resolution of antibiotic-LSU maps. a-e**, Fourier-shell correlation (FSC) curve of the focused refined LSU maps for (**a**) dataset 1, (**b**) dataset 2, (**c**) dataset 3, (**d**) dataset 4, and (**e**) dataset 5. The different curves include the masked map (green), unmasked map (blue) and the phase-randomized masked map (red). The dashed line at FSC 0.143 indicates that the average resolution for the LSU maps for datasets 1–5 are 1.86 Å, 1.83 Å, 1.64 Å, 1.98 Å, 1.65 Å, respectively. **f-j**, Transverse section of the LSU coloured by local resolution for the focused refined maps calculated from (**f**) dataset 1, (**g**) dataset 2, (**h**) dataset 3, (**i**) dataset 4, and (**j**) dataset 5.

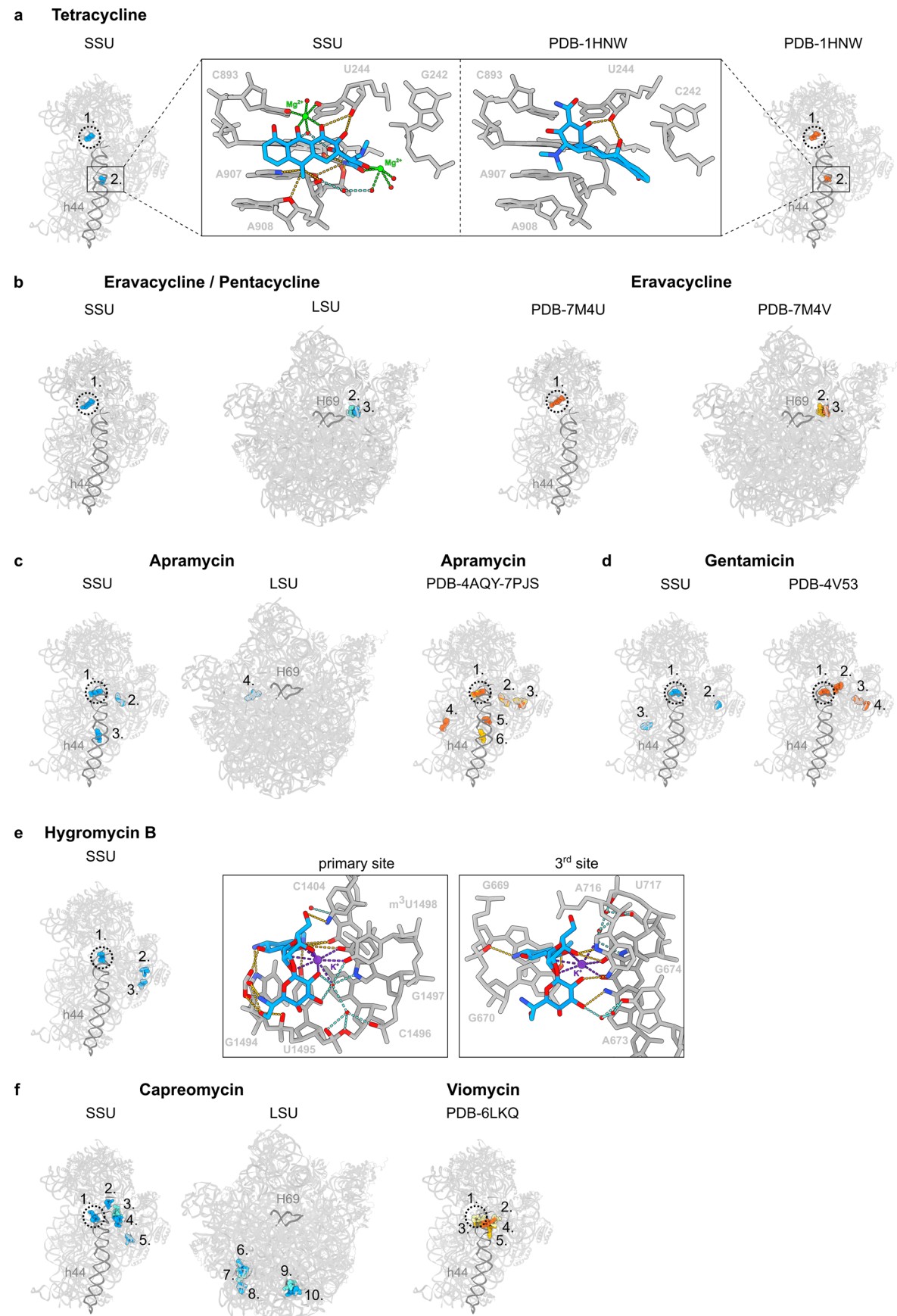

**Extended Data Fig. 5 | See next page for caption.**

**Extended Data Fig. 5 | Secondary binding sites of antibiotics on the ribosome.**
**a**, Overview of the primary (1.) and secondary (2.) binding sites of tetracycline (blue) on the SSU determined here (left) and tetracycline (red) reported previously on the SSU (right, PDB ID 1HNW)[26], with boxed zooms revealing the inverse orientation of tetracycline and distinct interactions observed with the rRNA. **b**, primary (1.) and secondary (2.-3.) binding sites of eravacycline/pentacycline (blue/cyan) on the SSU and LSU (left), compared with sites (red/orange) reported previously for eravacycline on the LSU (right)(PDB ID 7M4V)[31]. **c**, primary (1.) and secondary (2.-4.) binding sites of apramycin (blue) on the SSU and LSU (left), compared with sites reported previously on the LSU (right) (PDB ID 4AQY and 7PJS)[7,29]. **d**, primary (1.) and secondary (2.-3.) binding sites of gentamicin (blue) on the SSU (right), compared with sites reported previously for gentamicin (red/orange) on the SSU (right)(PDB ID 4V53))[28]. **e**, primary (1.) and secondary (2.-3.) binding sites of hygromycin B (blue) on the SSU (left), with insets showing the similarity in the coordination of a putative $K^+$ ion in the primary and 3rd site. **f**, primary (1.) and secondary (2.-4.) binding sites of capreomycin (blue) on the SSU and LSU (left), compared with sites reported previously for viomycin on the SSU (right)(PDB ID 6LKQ)[33].

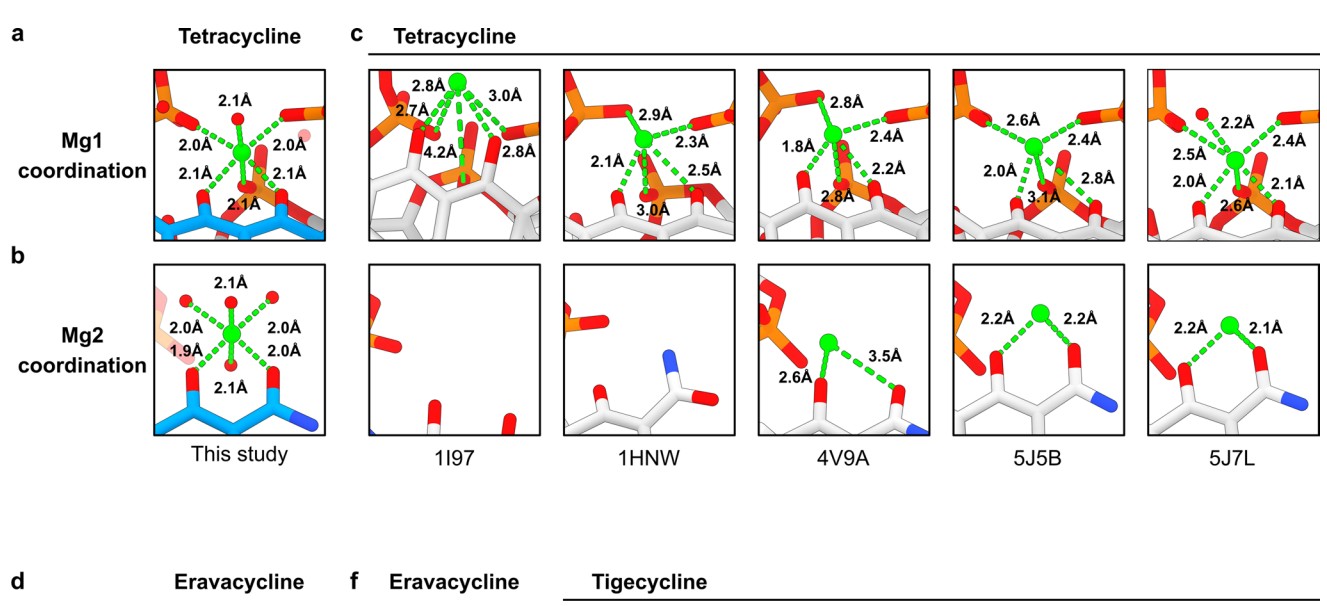

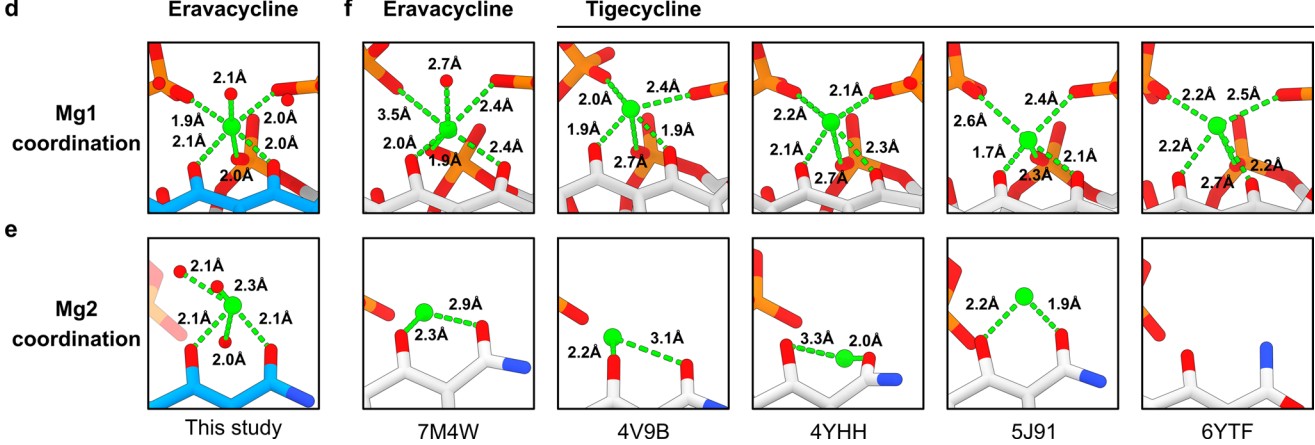

**Extended Data Fig. 6 | Coordination of magnesium ions by tetracyclines.**
**a-c**, Coordination of (**a**) Mg1 (green) and (**b**) Mg2 by tetracycline (blue) and waters (red balls) and phosphate-oxygens of 16S rRNA determined here, compared with (**c**) previous structures PDB ID 1I97[27], 1HNW[26], 4V9A[93], 5J5B[94] and 5J7L[94]. Distances are provided for the green dashed lines. **d-f**, Coordination of (**d**) Mg1 (green)

and (**e**) Mg2 by eravacycline (blue) and waters (red spheres) and phosphate-oxygens of 16S rRNA determined here, compared with (**f**) previous structures of eravacycline (PDB ID 7M4W)[31] and tigecycline (PDB ID 4V9B[93], 4YHH[100], 5J91[94] and 6YTF[101]). Distances are provided for the green dashed lines.

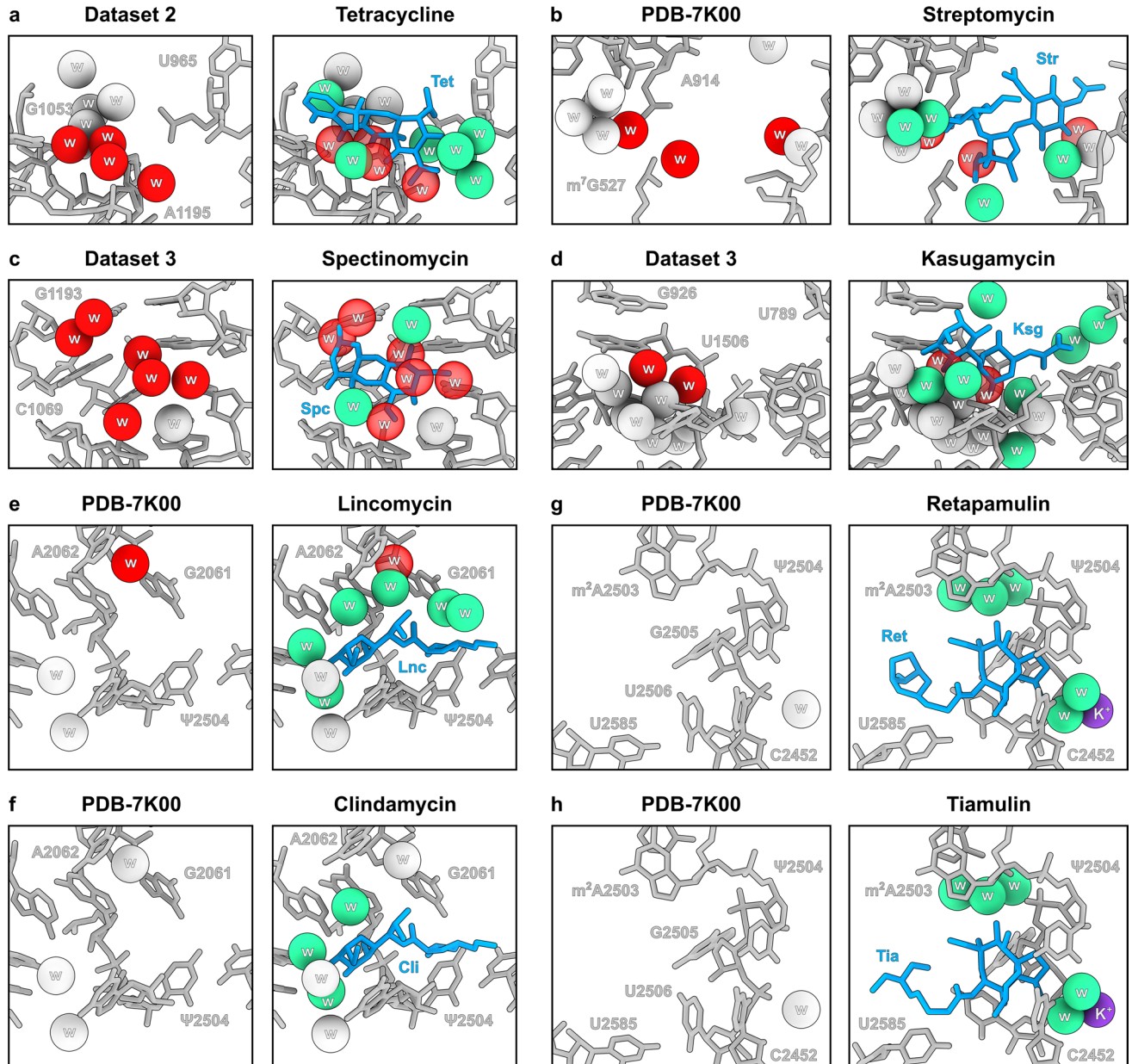

**Extended Data Fig. 7 | Displacement of waters upon antibiotic binding.**
**a**, View of tetracycline binding site (16S rRNA, grey) in the absence (left panel) and presence of tetracycline (right panel). Waters (spheres) that are displaced upon tetracycline (Tet, blue) binding are colored red. **b**, View of streptomycin binding site (16S rRNA, grey) in the absence (left panel) and presence of streptomycin (right panel). Waters (spheres) that are displaced upon streptomycin (Str, blue) binding are colored red. **c**, View of spectinomycin binding site (16S rRNA, grey) in the absence (left panel) and presence of spectinomycin (right panel). Waters (spheres) that are displaced upon spectinomycin (Spc, blue) binding are colored red. **d**, View of kasugamycin binding site (16S rRNA, grey) in the absence (left panel) and presence of kasugamycin (right panel). Waters (spheres) that are displaced upon kasugamycin (Ksg, blue) binding are colored red. **e-h**, View of (**e**) lincomycin, (**f**) clindamycin, (**g**) tiamulin and (**h**) retapamulin binding site (23S rRNA, grey) in the absence (left panel) and presence of the drug (right panel). Waters (spheres) that are displaced upon binding of the antibiotic are colored red. In (**a**)-(**h**), waters (spheres) that are stabilized upon drug binding are colored green.

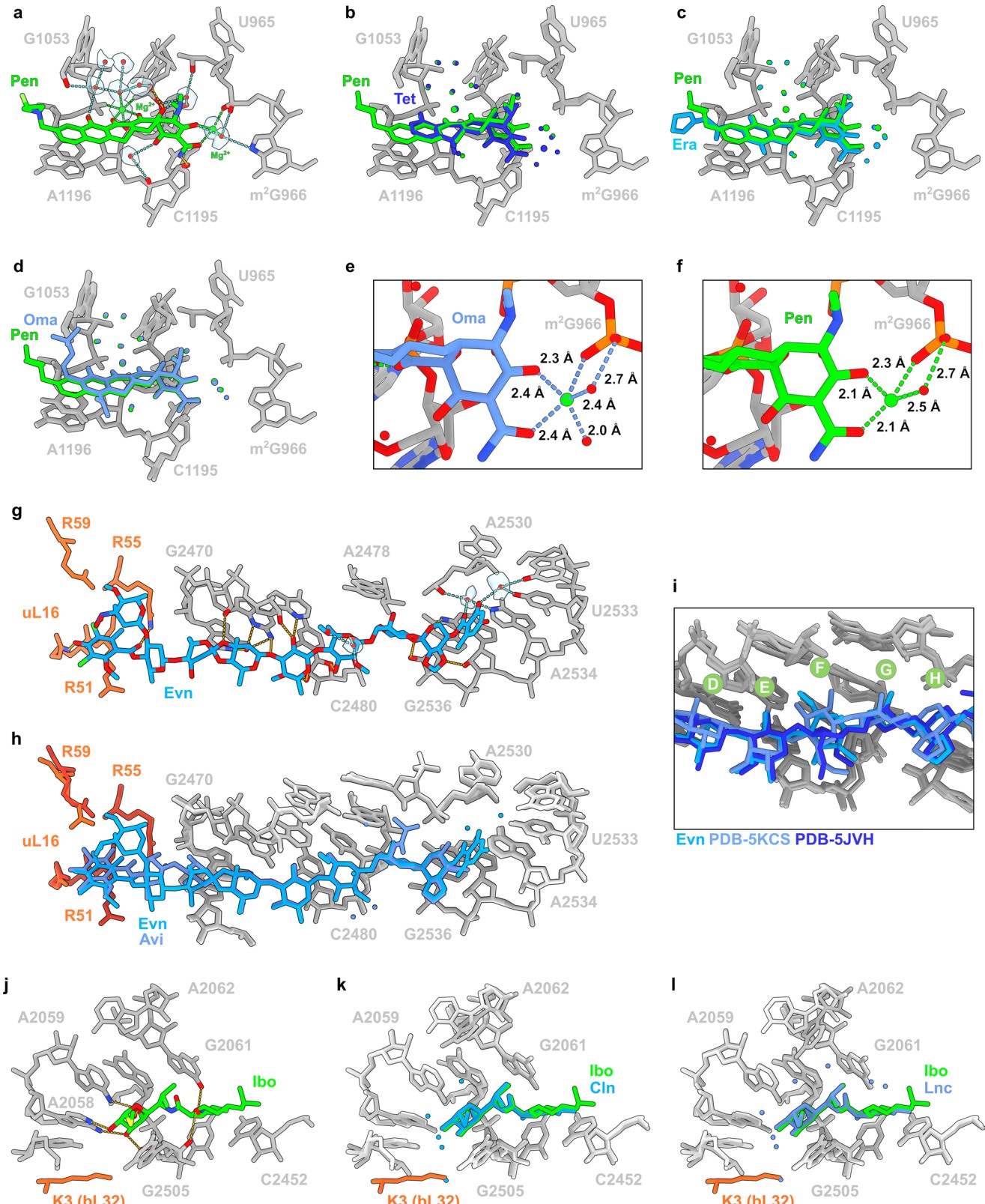

**Extended Data Fig 8 | Comparisons of pentacycline- and evernimicin-ribosome structures. a**, Interaction of pentacycline on the SSU head at 1.89 Å. **b-d**, Superimposition of pentacycline from (**a**) with (**b**) tetracycline on the SSU head determined at 1.8 Å (from Fig. 2), (**c**) eravacycline on the SSU head determined at 2.2 Å (from Fig. 4), and (**d**) omadacycline on the SSU head determined at 2.1 Å (from Fig. 4), **e-f**, Zoom of the coordination of the Mg2 for (**e**) omadacycline and (**f**) pentacycline. **g**, Interaction of evernimicin (Evn) on the

LSU at 1.9 Å. **h**, Superimposition of evernimicin (Evn) from (**g**) with avilamycin (Avi) (from Fig. 3). **i**, Superimposition of Evn (blue) from (**g**) with previous structures of Evn in complex with the ribosome (PDB ID 5KCS and 5JVH)[43,44]. **j**, Interaction of iboxamycin (green) on the 70S ribosome at 2.5 Å (PDB ID 7RQ8)[46]. **k-l**, superimposition of iboxamycin (green) from (**j**) with (**k**) clindamycin (light blue) and (**l**) lincomycin (dark blue) determined here at 2.0 Å and 1.6 Å, respectively.

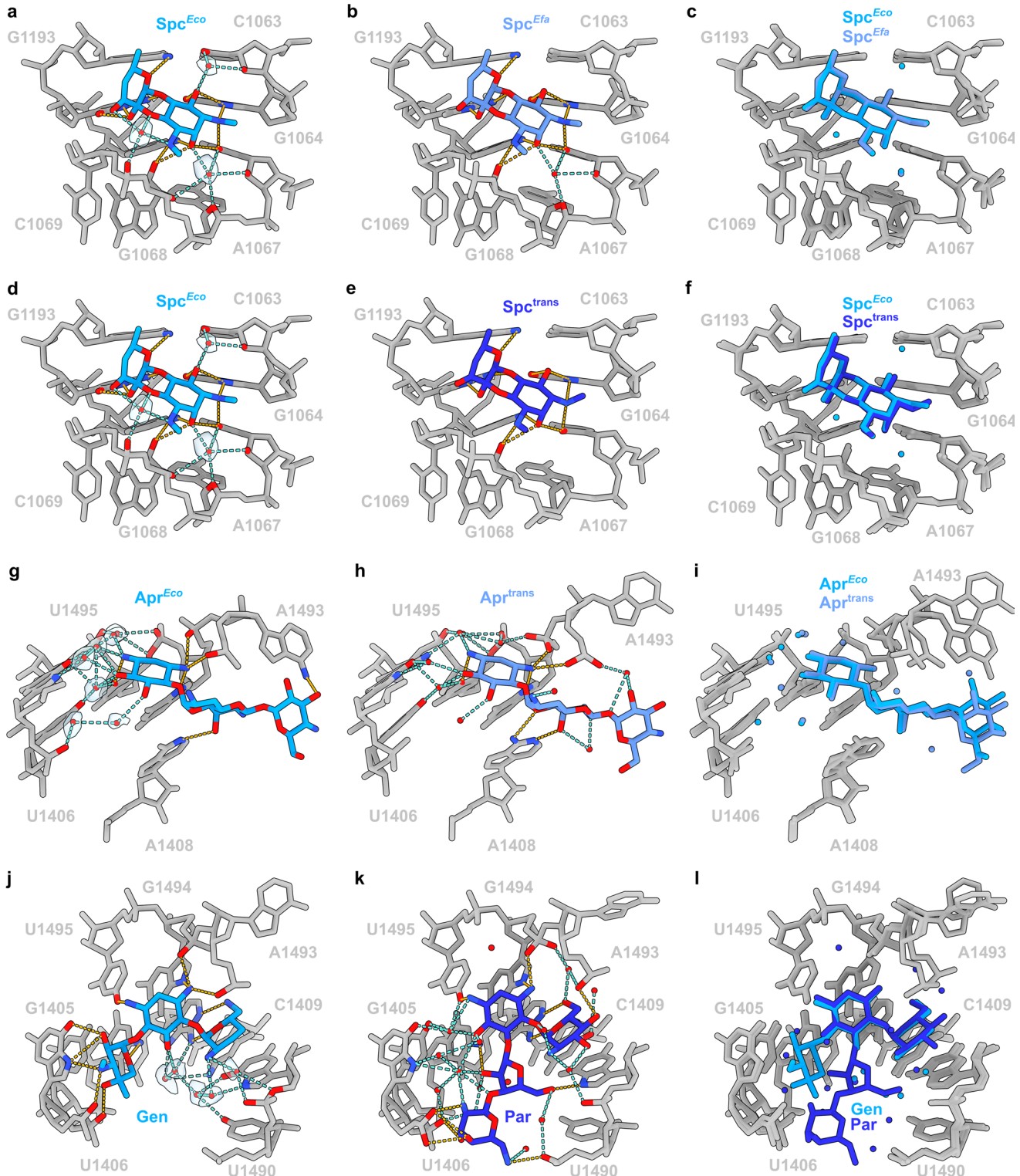

**Extended Data Fig. 9 | Comparisons of antibiotic-ribosome structures.**
**a-b**, Interaction of spectinomycin (Spc) determined (**a**) here on *E. coli* (Spc^*E.coli*)
at 1.8 Å (from Fig. 2) and (b) on the *E. faecalis* 70S ribosome (Spc^*E.fae.*) at 2.4 Å[8].
**c**, Superimposition of (**a**) and (**b**). **d-e**, Interaction of spectinomycin (Spc) deter-
mined (**d**) here on *E. coli* (Spc^*E.coli*) at 1.8 Å (from Fig. 2) and (**e**) within an *E. coli*
translocation intermate (Spc^*Trans*) at 2.54 Å[55]. **f**, Superimposition of (**d**) and (**e**).
**g-h**, Interaction of apramycin (Apr) determined (**g**) here on *E. coli* (Apr^*E.coli*)
(from Fig. 2) and (h) within an *E. coli* translocation intermediate (Apr^*Trans*) at
2.35 Å[7]. **i**, Superimposition of (**g**) and (**h**). **j-k**, Interaction of (**j**) gentamicin (Gen)
determined here on *E. coli* (Gen) (from Fig. 2) and (**k**) paromomycin (Par) on
an *E. coli* bearing mRNA, A- and P-site tRNAs at 1.98 Å[12]. **l**, Superimposition of
(**i**) and (**j**).

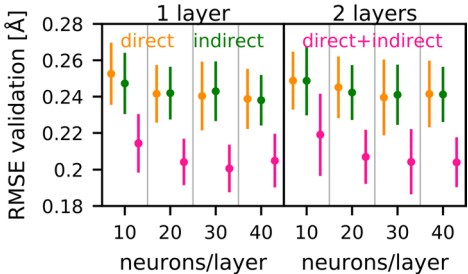

**Extended Data Fig. 10 | Testing of parameters of the neural network models.** For different numbers of layers and neurons per layer, the RMSE of predicted rmsd values relative to rmsd values obtained from the simulations for the cross-validation sets is shown. Mean (circles) and standard deviations (lines) obtained from training and cross-validation against 10 different training data sets. Colors denote different h-bond types used as input for the networks.

# Reporting Summary

## Statistics

For all statistical analyses, confirm that the following items are present in the figure legend, table legend, main text, or Methods section.

| n/a | Confirmed | |
|---|---|---|
| ☐ | ☒ | The exact sample size (*n*) for each experimental group/condition, given as a discrete number and unit of measurement |
| ☐ | ☒ | A statement on whether measurements were taken from distinct samples or whether the same sample was measured repeatedly |
| ☒ | ☐ | The statistical test(s) used AND whether they are one- or two-sided *Only common tests should be described solely by name; describe more complex techniques in the Methods section.* |
| ☒ | ☐ | A description of all covariates tested |
| ☐ | ☒ | A description of any assumptions or corrections, such as tests of normality and adjustment for multiple comparisons |
| ☐ | ☒ | A full description of the statistical parameters including central tendency (e.g. means) or other basic estimates (e.g. regression coefficient) AND variation (e.g. standard deviation) or associated estimates of uncertainty (e.g. confidence intervals) |
| ☒ | ☐ | For null hypothesis testing, the test statistic (e.g. *F*, *t*, *r*) with confidence intervals, effect sizes, degrees of freedom and *P* value noted *Give P values as exact values whenever suitable.* |
| ☐ | ☒ | For Bayesian analysis, information on the choice of priors and Markov chain Monte Carlo settings |
| ☒ | ☐ | For hierarchical and complex designs, identification of the appropriate level for tests and full reporting of outcomes |
| ☒ | ☐ | Estimates of effect sizes (e.g. Cohen's *d*, Pearson's *r*), indicating how they were calculated |

*Our web collection on statistics for biologists contains articles on many of the points above.*

## Software and code

Policy information about availability of computer code

| Data collection | CryoEM data were collected using the EPU 3.0 software (FEI, Netherlands) or SerialEM 3.8. |
|---|---|
| Data analysis | RELION v 3.1 or 4 with MotionCor2 v1.2.1, CTFFIND 4.1.14, and crYOLO 1.8.0b47 or Topaz within Relion v4 were used for processing micrographs, picking particles, classification and refining cryo-EM maps. Relion v4 was used to calculate local resolution. Coot v0.9.8.1 for model building and ServalCat v0.3.1 with REFMAC 5 v5.8.0415 for model refinement and statistics, with structural restraints generated by PRODRG2, aceDRG, or Phenix eLBOW. Figures were generated using ChimeraX v1.3 and v1.6.1. Molecular dynamics simulations was performed in GROMACS-2022.4. |

For manuscripts utilizing custom algorithms or software that are central to the research but not yet described in published literature, software must be made available to editors and reviewers. We strongly encourage code deposition in a community repository (e.g. GitHub). See the Nature Portfolio guidelines for submitting code & software for further information.

## Data

Policy information about availability of data

All manuscripts must include a data availability statement. This statement should provide the following information, where applicable:

- Accession codes, unique identifiers, or web links for publicly available datasets
- A description of any restrictions on data availability
- For clinical datasets or third party data, please ensure that the statement adheres to our policy

Initial models for structure were generated based on the available molecular model of the E. coli 70S ribosome at 1.98 Å (PDB ID 7K00) and potassium ions were designated according to previous models (PDB-6QNQ). The cryo-electron microscopy maps for the antibiotic-ribosome complexes have been deposited in the EMDataBank with the accession code EMD-16520 (Dataset 1, SSU-head), EMD-16526 (Dataset 1, SSU-body), EMD-16530 (Dataset 1, LSU), EMD-16536 (Dataset 2,

SSU-head), EMD-16612 (Dataset 2, SSU-body), EMD-16613 (Dataset 2, LSU), EMD-16615 (Dataset 3, SSU-head), EMD-16645 (Dataset 3, SSU-body), EMD-16646 (Dataset 3, LSU), EMD-16620 (Dataset 4, SSU-head), EMD-16650 (Dataset 4, SSU-body), EMD-16641 (Dataset 4, LSU), EMD-16644 (Dataset 5, SSU-head), EMD-16651 (Dataset 5, SSU-body) and EMD-16652 (Dataset 5, LSU). The respective coordinates for electron-microscopy-based model of the antibiotic-ribosome complexes are deposited in the ProteinDataBank with the accession code PDB 8CA7 (Dataset 1, SSU-head), PDB 8CAI (Dataset 1, SSU-body), PDB 8CAM (Dataset 1, LSU), PDB 8CAZ (Dataset 2, SSU-head), PDB 8CEP (Dataset 2, SSU-body), PDB 8CEU (Dataset 2, LSU), PDB 8CF1 (Dataset 3, SSU-head), PDB 8CGJ (Dataset 3, SSU-body), PDB 8CGK (Dataset 3, LSU), PDB 8CF8 (Dataset 4, SSU-head), PDB 8CGR (Dataset 4, SSU-body), PDB 8CGD (Dataset 4, LSU), PDB 8CGI (Dataset 5, SSU-head), PDB 8CGU (Dataset 5, SSU-body) and PDB 8CGV (Dataset 5, LSU).

# Field-specific reporting

Please select the one below that is the best fit for your research. If you are not sure, read the appropriate sections before making your selection.

☒ Life sciences ☐ Behavioural & social sciences ☐ Ecological, evolutionary & environmental sciences

For a reference copy of the document with all sections, see nature.com/documents/nr-reporting-summary-flat.pdf

# Life sciences study design

All studies must disclose on these points even when the disclosure is negative.

| | |
|---|---|
| Sample size | no sample size calculation was performed. The sample size was selected on the basis of a three-day data collection, which was chosen to obtain sufficient number of particles to bring the resolution of the resulting complexes towards or below 2A resolution. |
| Data exclusions | Micrographs with low estimated resolution or poorly fitted CTFs were discarded, as were particles that clustered into poorly defined classes during 2D and 3D classification. |
| Replication | The structure of streptomycin was performed in duplicate and the similarity in the obtained molecular models indicates that the replication was successful. Other remaining antibiotic structures were not replicated, although in many cases the antibiotics had similar scaffolds and the results indicated that their interactions were successfully replicated. |
| Randomization | For 3D refinement in RELION, particles are randomly placed in one of two subsets. These subsets are maintained for CTF refinement. Otherwise, no randomization was performed. For the molecular dynamics simulations, to obtain statistical uncertainties, 1000 subsets of conformations were randomly selected and the analysis was repeated on each subset. |
| Blinding | No blinding was performed as blinding is not possible or not applicable for the experiments because the identity of the analyzed sample was known |

# Reporting for specific materials, systems and methods

We require information from authors about some types of materials, experimental systems and methods used in many studies. Here, indicate whether each material, system or method listed is relevant to your study. If you are not sure if a list item applies to your research, read the appropriate section before selecting a response.

## Materials & experimental systems

| n/a | Involved in the study |
|---|---|
| ☒ ☐ | Antibodies |
| ☒ ☐ | Eukaryotic cell lines |
| ☒ ☐ | Palaeontology and archaeology |
| ☒ ☐ | Animals and other organisms |
| ☒ ☐ | Human research participants |
| ☒ ☐ | Clinical data |
| ☒ ☐ | Dual use research of concern |

## Methods

| n/a | Involved in the study |
|---|---|
| ☒ ☐ | ChIP-seq |
| ☒ ☐ | Flow cytometry |
| ☒ ☐ | MRI-based neuroimaging |

# Antibodies

| | |
|---|---|
| Antibodies used | N/A |
| Validation | *Describe the validation of each primary antibody for the species and application, noting any validation statements on the manufacturer's website, relevant citations, antibody profiles in online databases, or data provided in the manuscript.* |

# Eukaryotic cell lines

Policy information about cell lines

| | |
|---|---|
| Cell line source(s) | N/A |

| Authentication | *Describe the authentication procedures for each cell line used OR declare that none of the cell lines used were authenticated.* |
| --- | --- |
| Mycoplasma contamination | *Confirm that all cell lines tested negative for mycoplasma contamination OR describe the results of the testing for mycoplasma contamination OR declare that the cell lines were not tested for mycoplasma contamination.* |
| Commonly misidentified lines (See ICLAC register) | *Name any commonly misidentified cell lines used in the study and provide a rationale for their use.* |

# Palaeontology and Archaeology

| Specimen provenance | N/A |
| --- | --- |
| Specimen deposition | *Indicate where the specimens have been deposited to permit free access by other researchers.* |
| Dating methods | *If new dates are provided, describe how they were obtained (e.g. collection, storage, sample pretreatment and measurement), where they were obtained (i.e. lab name), the calibration program and the protocol for quality assurance OR state that no new dates are provided.* |

☐ Tick this box to confirm that the raw and calibrated dates are available in the paper or in Supplementary Information.

| Ethics oversight | *Identify the organization(s) that approved or provided guidance on the study protocol, OR state that no ethical approval or guidance was required and explain why not.* |
| --- | --- |

Note that full information on the approval of the study protocol must also be provided in the manuscript.

# Animals and other organisms

Policy information about studies involving animals; ARRIVE guidelines recommended for reporting animal research

| Laboratory animals | N/A |
| --- | --- |
| Wild animals | *Provide details on animals observed in or captured in the field; report species, sex and age where possible. Describe how animals were caught and transported and what happened to captive animals after the study (if killed, explain why and describe method; if released, say where and when) OR state that the study did not involve wild animals.* |
| Field-collected samples | *For laboratory work with field-collected samples, describe all relevant parameters such as housing, maintenance, temperature, photoperiod and end-of-experiment protocol OR state that the study did not involve samples collected from the field.* |
| Ethics oversight | *Identify the organization(s) that approved or provided guidance on the study protocol, OR state that no ethical approval or guidance was required and explain why not.* |

Note that full information on the approval of the study protocol must also be provided in the manuscript.

# Human research participants

Policy information about studies involving human research participants

| Population characteristics | N/A |
| --- | --- |
| Recruitment | *Describe how participants were recruited. Outline any potential self-selection bias or other biases that may be present and how these are likely to impact results.* |
| Ethics oversight | *Identify the organization(s) that approved the study protocol.* |

Note that full information on the approval of the study protocol must also be provided in the manuscript.

# Clinical data

Policy information about clinical studies

All manuscripts should comply with the ICMJE guidelines for publication of clinical research and a completed CONSORT checklist must be included with all submissions.

| Clinical trial registration | N/A |
| --- | --- |
| Study protocol | *Note where the full trial protocol can be accessed OR if not available, explain why.* |
| Data collection | *Describe the settings and locales of data collection, noting the time periods of recruitment and data collection.* |
| Outcomes | *Describe how you pre-defined primary and secondary outcome measures and how you assessed these measures.* |

# Dual use research of concern

Policy information about dual use research of concern

## Hazards

Could the accidental, deliberate or reckless misuse of agents or technologies generated in the work, or the application of information presented in the manuscript, pose a threat to:

| No | Yes | |
|----|-----|---|
| ☒ | ☐ | Public health |
| ☒ | ☐ | National security |
| ☒ | ☐ | Crops and/or livestock |
| ☒ | ☐ | Ecosystems |
| ☒ | ☐ | Any other significant area |

## Experiments of concern

Does the work involve any of these experiments of concern:

| No | Yes | |
|----|-----|---|
| ☒ | ☐ | Demonstrate how to render a vaccine ineffective |
| ☒ | ☐ | Confer resistance to therapeutically useful antibiotics or antiviral agents |
| ☒ | ☐ | Enhance the virulence of a pathogen or render a nonpathogen virulent |
| ☒ | ☐ | Increase transmissibility of a pathogen |
| ☒ | ☐ | Alter the host range of a pathogen |
| ☒ | ☐ | Enable evasion of diagnostic/detection modalities |
| ☒ | ☐ | Enable the weaponization of a biological agent or toxin |
| ☒ | ☐ | Any other potentially harmful combination of experiments and agents |

# ChIP-seq

## Data deposition

☐ Confirm that both raw and final processed data have been deposited in a public database such as GEO.

☐ Confirm that you have deposited or provided access to graph files (e.g. BED files) for the called peaks.

| | |
|---|---|
| **Data access links** *May remain private before publication.* | N/A |
| **Files in database submission** | *Provide a list of all files available in the database submission.* |
| **Genome browser session** (e.g. UCSC) | *Provide a link to an anonymized genome browser session for "Initial submission" and "Revised version" documents only, to enable peer review.  Write "no longer applicable" for "Final submission" documents.* |

## Methodology

| | |
|---|---|
| **Replicates** | *Describe the experimental replicates, specifying number, type and replicate agreement.* |
| **Sequencing depth** | *Describe the sequencing depth for each experiment, providing the total number of reads, uniquely mapped reads, length of reads and whether they were paired- or single-end.* |
| **Antibodies** | *Describe the antibodies used for the ChIP-seq experiments; as applicable, provide supplier name, catalog number, clone name, and lot number.* |
| **Peak calling parameters** | *Specify the command line program and parameters used for read mapping and peak calling, including the ChIP, control and index files used.* |
| **Data quality** | *Describe the methods used to ensure data quality in full detail, including how many peaks are at FDR 5% and above 5-fold enrichment.* |
| **Software** | *Describe the software used to collect and analyze the ChIP-seq data. For custom code that has been deposited into a community repository, provide accession details.* |

# Flow Cytometry

## Plots

Confirm that:

☐ The axis labels state the marker and fluorochrome used (e.g. CD4-FITC).

☐ The axis scales are clearly visible. Include numbers along axes only for bottom left plot of group (a 'group' is an analysis of identical markers).

☐ All plots are contour plots with outliers or pseudocolor plots.

☐ A numerical value for number of cells or percentage (with statistics) is provided.

## Methodology

| | |
|---|---|
| Sample preparation | N/A |
| Instrument | *Identify the instrument used for data collection, specifying make and model number.* |
| Software | *Describe the software used to collect and analyze the flow cytometry data. For custom code that has been deposited into a community repository, provide accession details.* |
| Cell population abundance | *Describe the abundance of the relevant cell populations within post-sort fractions, providing details on the purity of the samples and how it was determined.* |
| Gating strategy | *Describe the gating strategy used for all relevant experiments, specifying the preliminary FSC/SSC gates of the starting cell population, indicating where boundaries between "positive" and "negative" staining cell populations are defined.* |

☐ Tick this box to confirm that a figure exemplifying the gating strategy is provided in the Supplementary Information.

# Magnetic resonance imaging

## Experimental design

| | |
|---|---|
| Design type | N/A |
| Design specifications | *Specify the number of blocks, trials or experimental units per session and/or subject, and specify the length of each trial or block (if trials are blocked) and interval between trials.* |
| Behavioral performance measures | *State number and/or type of variables recorded (e.g. correct button press, response time) and what statistics were used to establish that the subjects were performing the task as expected (e.g. mean, range, and/or standard deviation across subjects).* |

## Acquisition

| | |
|---|---|
| Imaging type(s) | *Specify: functional, structural, diffusion, perfusion.* |
| Field strength | *Specify in Tesla* |
| Sequence & imaging parameters | *Specify the pulse sequence type (gradient echo, spin echo, etc.), imaging type (EPI, spiral, etc.), field of view, matrix size, slice thickness, orientation and TE/TR/flip angle.* |
| Area of acquisition | *State whether a whole brain scan was used OR define the area of acquisition, describing how the region was determined.* |

Diffusion MRI       ☐ Used       ☐ Not used

## Preprocessing

| | |
|---|---|
| Preprocessing software | *Provide detail on software version and revision number and on specific parameters (model/functions, brain extraction, segmentation, smoothing kernel size, etc.).* |
| Normalization | *If data were normalized/standardized, describe the approach(es): specify linear or non-linear and define image types used for transformation OR indicate that data were not normalized and explain rationale for lack of normalization.* |
| Normalization template | *Describe the template used for normalization/transformation, specifying subject space or group standardized space (e.g. original Talairach, MNI305, ICBM152) OR indicate that the data were not normalized.* |
| Noise and artifact removal | *Describe your procedure(s) for artifact and structured noise removal, specifying motion parameters, tissue signals and physiological signals (heart rate, respiration).* |

| Volume censoring | *Define your software and/or method and criteria for volume censoring, and state the extent of such censoring.* |

## Statistical modeling & inference

| Model type and settings | *Specify type (mass univariate, multivariate, RSA, predictive, etc.) and describe essential details of the model at the first and second levels (e.g. fixed, random or mixed effects; drift or auto-correlation).* |
| Effect(s) tested | *Define precise effect in terms of the task or stimulus conditions instead of psychological concepts and indicate whether ANOVA or factorial designs were used.* |

Specify type of analysis: ☐ Whole brain ☐ ROI-based ☐ Both

| Statistic type for inference<br>(See Eklund et al. 2016) | *Specify voxel-wise or cluster-wise and report all relevant parameters for cluster-wise methods.* |
| Correction | *Describe the type of correction and how it is obtained for multiple comparisons (e.g. FWE, FDR, permutation or Monte Carlo).* |

## Models & analysis

| n/a | Involved in the study |
| --- | --- |
| ☐ | ☐ Functional and/or effective connectivity |
| ☐ | ☐ Graph analysis |
| ☐ | ☐ Multivariate modeling or predictive analysis |

| Functional and/or effective connectivity | *Report the measures of dependence used and the model details (e.g. Pearson correlation, partial correlation, mutual information).* |
| Graph analysis | *Report the dependent variable and connectivity measure, specifying weighted graph or binarized graph, subject- or group-level, and the global and/or node summaries used (e.g. clustering coefficient, efficiency, etc.).* |
| Multivariate modeling and predictive analysis | *Specify independent variables, features extraction and dimension reduction, model, training and evaluation metrics.* |

