## [Peer Review File · Nature Structural & Molecular Biology]

Peer Review Information

Manuscript Title: Structural conservation of antibiotic interaction with ribosomes

Corresponding author name(s): Daniel Wilson

Reviewer Comments & Decisions:

Decision Letter, initial version:

Message: 12th Apr 2023

Dear Prof. Wilson,

Thank you again for submitting your manuscript "Structural conservation of antibiotic interaction with ribosomes". We now have comments (below) from the 3 reviewers who evaluated your paper and we have editorially discussed their input amongst the whole editorial team. In light of those reports and our discussions, we remain interested in your study and would like to see your response to the comments of the referees, in the form of a revised manuscript.

You will see that all referees appreciate the high-resolution of the structures and their potential implication in the refinement of existing or design of new ribosome inhibitors/antibiotics. There are, however, a few important issues and suggestions that should be addressed in a revision. From a technical standpoint, we request that you address the concern of both reviewer #2 (R#2) and reviewer #3 (R#3) with respect to the very high antibiotic concentrations used for certain antibiotics and their potential artefactual effects. In addition, we would request that you please perform and provide additional repeats for certain inhibitors/antibiotics to be able to unambiguously assign water molecules and ions, in accordance to the guidance of R#2 (point 1). We editorially agree with point 2 of R#2 that any biochemical experiments supporting the existence of potassium ions in the HygB-ribosome complex would further validate the value of this work. Finally, we ask you to please format/clarify/simplify current figures and text in accordance with the guidance of all the reviewers to help the reader. As always, please be sure to address/respond to all concerns of the referees in full in a point-by-point response and highlight all changes in the revised manuscript text file. If you have comments that are intended for editors only, please include those in a separate cover letter.

We expect to see your revised manuscript within 3 months. If you cannot send it within this time, please contact us to discuss an extension; we would still consider your revision, provided that no similar work has been accepted for publication at NSMB or published elsewhere.

Reporting Summary:

When submitting the revised version of your manuscript, please pay close attention to our [href="https://www.nature.com/nature-portfolio/editorial-policies/image-integrity">Digital Image Integrity Guidelines. and to the following points below:](https://www.nature.com/nature-portfolio/editorial-policies/image-integrity)

Data availability: this journal strongly supports public availability of data. All data used in accepted papers should be available via a public data repository, or alternatively, as

Supplementary Information. If data can only be shared on request, please explain why in your Data Availability Statement, and also in the correspondence with your editor. Please note that for some data types, deposition in a public repository is mandatory - more information on our data deposition policies and available repositories can be found below: <https://www.nature.com/nature-research/editorial-policies/reporting-standards#availability-of-data>

IMPORTANT: We require deposition of coordinates (and, in the case of crystal structures, structure factors) into the Protein Data Bank with the designation of immediate release upon publication (HPUB). Electron microscopy-derived density maps and coordinate data must be deposited in EMDB and released upon publication. Failure to provide such maps in their latest format, as well as our corresponding table containing summary statistics is the most common cause for delays at the accept stage. To avoid delays in publication, dataset accession numbers must be supplied with the final accepted manuscript and appropriate release dates must be indicated at the galley proof stage.

[Redacted]

Sincerely,

Dimitris Typas
Associate Editor
Nature Structural & Molecular Biology
ORCID: 0000-0002-8737-1319

Referee expertise:

Referee #1: Cryo-EM, ribosomes, antibiotics

Referee #2: Structural biology, ribosomes, antibiotics

Referee #3: Cryo-EM, ribosomes, antibiotics

Reviewers' Comments:

Reviewer #1:

Remarks to the Author:

In this manuscript, authors describe high-resolution (1.6-2.2Å) structures of 17 different compounds from 6 distinct antibiotic classes bound to the bacterial ribosome. These improved resolutions provide the most precise description of compound-ribosome interactions to date, encompassing solvent networks that mediate multiple additional interactions between the drugs and their target. Particularly notable is the unambiguous detection of ordered water molecules that mediate interactions between the drugs and the ribosome, and in some instances presence of the second layer of water. The high solvation of antibiotics observed in this work is a novel finding, one enabled by the high resolution of the structures. The relevance of solvation to antibiotic binding is strengthened by MD simulations on lincomycin bound complex that suggests that waters involved in interactions with the antibiotic appear stable, suggesting its importance in physiological antibiotic binding. While the structural analysis used vacant ribosomes lacking m- and t-RNAs, a comparison of antibiotic-bound structures obtained in this work with those of antibiotic-bound ribosomes trapped in various functional states containing m- and t-RNAs shows that positioning and interaction of antibiotics with the ribosome are highly similar. These are unprecedented insights into antibiotic-ribosome interactions. High-resolution structures describing metal ion and ordered water interactions provided in this work are necessary to drive the design of antibiotics. Furthermore, high-resolution structural data provided in this work may enable the design of RNA targeting therapeutics in other well-defined RNA binding pockets, overcoming currently limited information of the role of water in RNA-ligand interactions.

One minor change is suggested in the abstract: Change RNA-based (a term commonly used for mRNA vaccines and similar modalities) to RNA-targeting therapies

Overall, this is a tour de force in the investigation of ribosome-bound antibiotic structures that will enable the design of new ribosome-targeting drugs.

Reviewer #2:

Remarks to the Author:

In the manuscript 'Structural conservation of antibiotic interaction' the authors present structures of 17 already known compounds with improved resolution. The authors chose the E.coli ribosome as a model to determine binding modes of compounds effective against wide spectra of pathogens. The paper gives a type of atlas for the interactions of the antibiotics with their binding pockets, being an excellent supplement to previous studies.

Indeed, not so long ago, the study of ribosome-binding inhibitors by cryo-electron microscopy was questioned because of insufficient resolution, the lack of a difference map

and overall underdevelopment. However, the method's upgrade has allowed inhibitors to be examined at a reasonably good resolution for several years now, and compounds have been repeatedly studied in complexes with prokaryotic and eukaryotic ribosomes. Of course, the resolution and quality of the electron map obtained is encouraging, but the fact that cryo-em is used to study inhibitors is not a new result. The pursuit of high resolution is admirable, but there will always be uncertainties and inaccuracies, especially in the identification of ions and ligands, as the method itself is not a direct method of distance determination.

Structures with all of the inhibitors studied have been obtained before, albeit at a lower resolution. The interactions found are almost identical to those that were found before. As a consequence, this article is mainly of scientific interest for pharmaceutical research.

Reviewer #3:

Remarks to the Author:

The work by Paternoga H. et al. "Structural conservation of antibiotic interaction with ribosomes" presents five high-resolution cryo-EM structures of the E. coli 70S ribosome in complex with antibiotics. Using the clever approach of including several compounds per sample, five datasets we collected to visualize 17 antibiotics. The analysis reports on the ribosome-binding principles of six antibiotic classes, tetracyclines, aminoglycosides, tuberactinomycins, orthosomycins, pleuromutilins, and lincosamides. The structures reveal that antibiotics that belong to the same class use a common binding mode. The high-resolution of the structures (1.6 to 2.2 Å) allows to visualize the antibiotic binding sites and to reliably position ions and water molecules that stabilize the antibiotics. The major advance of this work is not only in establishing the unambiguous conformation of the various antibiotics, clearing out ambiguities of antibiotic binding modes on ribosomes from past structures due to limited resolution, but also in elucidating the precise interaction network mediated by ordered ions and water molecules. The solvated structure with the highest resolution, that of the lincomycin-bound ribosome, was further used in molecular dynamic simulations to establish the relevance of the water-mediated interactions at higher temperatures, showing that the interaction network is maintained despite higher fluctuations.

This is a highly descriptive study. However, the unprecedented details reported in this manuscript of the interactions between clinically relevant drugs and the ribosome allow to distinguish water molecules that become ordered upon drug binding and/or are pre-ordered prior to drug binding, from those that are displaced by the drugs. The data is expected to provide conceivable avenues to be explored for the design of improved ribosome inhibitors. The following points should be addressed prior to publication.

The observation of secondary binding sites for many of the studied antibiotics should be discussed considering the high concentrations of antibiotics used for EM. For instance, hygromycin B was used at a final concentration of 200 μM while capreomycin, which is observed at nine secondary sites, was used at 100 μM . Similarly, the concentration of ribosomes used during complex formation should be explicitly written in the Methods. For grid preparation, the authors used antibiotic-ribosome complexes at 7 OD/mL, which appears to be $\sim 0.2 \mu\text{M}$. Is this a dilution of the ribosome-antibiotic complexes? The apparent large excess of drugs over ribosomes (~ 500 - 1000 -fold) likely explains most of the secondary binding sites observed, which should be mentioned.

Fig. 2

It would be helpful to the reader to label nucleotides. Some are labeled, while some are not (i.e. streptomycin, gentamicin, apramycin, and kasugamycin binding sites).

Fig. 2 Spectinomycin

In the text on page 8, "...where two waters are coordinated by the oxygens (O7) located in the nucleobases of C1063 and C1066..." Based on the figure, it appears it is the oxygen O2 in C1063 and C1066 that interacts with the waters.

Fig. 3 Lincomycin

Label the 7OH group in the panel

Fig. 4

Many of the panels are overwhelming. One that is particularly difficult to grasp is 4g. Try alternate orientations? Panel 4j: Interactions with the backbone of G2505 are unclear. With all of the red oxygens, blue nitrogens, and orange phosphates, it makes the panels heavy. Coloring only the groups involved in interactions may help.

Page 14

"...we note that a striking similarity between the..." we note that there is a striking... or we note a striking...

"...revealing that all waters molecules remained stably..." delete 's' at water

"Indeed, our neural network analysis suggest that both..." suggests

Ext. Data Fig. 2, 3, and 4

The FSC graphs should be labeled "Dataset 1" etc. at the top of each graph (like in the legend).

In the legend, define the blue, green, and red FSC curves.

Same comment for Extended Data Fig. 7. In panels a, c, and d, replace "Round 2" and "Round 3" with dataset number.

Author Rebuttal to Initial comments

Decision Letter, first revision:

Message: Our ref: NSMB-A47395A

11th May 2023

Dear Prof. Wilson,

Thank you for submitting your revised manuscript "Structural conservation of antibiotic interaction with ribosomes" (NSMB-A47395A). It has now been seen by the original referees and their comments are below. The reviewers find that the paper has improved in revision, and therefore we'll be happy in principle to publish it in Nature Structural & Molecular Biology, pending minor revisions to satisfy reviewer #3's final requests about atom numbering and updating the colouring scheme at certain points to help the reader. Furthermore, as always, minor changes will be necessary to comply with our editorial and formatting guidelines.

To facilitate our work at this stage, it is important that we have a copy of the main text as a word file. If you could please send along a word version of this file as soon as possible, we would greatly appreciate it; please make sure to copy the NSMB account (cc'ed above).

Sincerely,

Dimitris Typas
Associate Editor
Nature Structural & Molecular Biology
ORCID: 0000-0002-8737-1319

Reviewer #1 (Remarks to the Author):

The response addresses concerns raised by the reviewers. The authors have clarified the likely origins of secondary binding sites and outlined the evidence used for the assignments of metal ions and waters, the relevance of which is strongly supported by the MD simulations as well as the conservation across related antibiotics from the same family. Other requested revisions to data presentation and interpretation have also been made. While some differences remain (eg, length of hydrogen bond), PDB files will allow future readers to decide on their own how much weight they want to put on these interactions.

Reviewer #2 (Remarks to the Author):

The authors have addressed all the questions that arose during the first reading. Overall, the article merits publication in the NSMB journal.

Reviewer #3 (Remarks to the Author):

The authors have addressed most of my inquiries. There are a few minor issues that need further attention, including the numbering of atoms of nucleotides C1063 and C1066 in Fig. 2 (Spectinomycin) and the coloring scheme of a few figure panels (see comments).

Original comment

Fig. 2 Spectinomycin

In the text on page 8, "...where two waters are coordinated by the oxygens (O7) located in the nucleobases of C1063 and C1066..." Based on the figure, it appears it is the oxygen O2 in C1063 and C1066 that interacts with the waters.

Authors' response

Both the O7 and O2 of C1063 and C1066 coordinate water molecules...this is seen in Figure 2 and stated in the text on page 8 "where two waters are coordinated by the oxygens (O7) located in the nucleobases of C1063 and C1066, together with their respective ribose 2' oxygens (Fig. 2)."

The authors should use the standard atom numbering in nucleotides. There is no oxygen O7 in the nucleobase of cytosine. The exocyclic oxygen in C and U is O2, while in the ribose, they are O2', O3', O4', and O5'. For reference, please see <https://www2.tulane.edu/~biochem/nolan/lectures/rna/frames/nucs.htm>

Original comment

Fig. 4

Many of the panels are overwhelming. One that is particularly difficult to grasp is 4g. Try alternate orientations?

Authors' response

Unfortunately, it is very difficult to find a single view where all water molecules are observed. We realize that 4g is rather complicated but it's the same view as 4h and 4i, and the only view where one can nicely see the overlap between the waters in these comparisons. An alternative view in 4g will disconnect it from 4h and 4i. Nevertheless, we have taken the reviewers advice from the comment below and with a reduced atom colouring focussing on the interactions, we hope that this makes the presentation less overwhelming.

Panel 4g is now clearer. I suggest to attempt a similar coloring scheme for panels 4a, 4d, and 4e. It would improve the clarity of the figure.

Similarly, using the same coloring strategy for the insets in the Ext. Data Fig. 5, in Ext.

Data Fig. 8a, g, and j, and in Ext. Data Fig. 9a-b, d-e, g-h, j-k will also improve these figures.

Author Rebuttal, first revision:

Reviewer #1:

Remarks to the Author:

The response addresses concerns raised by the reviewers. The authors have clarified the likely origins of secondary binding sites and outlined the evidence used for the assignments of metal ions and waters, the relevance of which is strongly supported by the MD simulations as well as the conservation across related antibiotics from the same family. Other requested revisions to data presentation and interpretation have also been made. While some differences remain (eg, length of hydrogen bond), PDB files will allow future readers to decide on their own how much weight they want to put on these interactions.

Reviewer #2:

Remarks to the Author:

The authors have addressed all the questions that arose during the first reading. Overall, the article merits publication in the NSMB journal.

Reviewer #3:

Remarks to the Author:

The authors have addressed most of my inquiries. There are a few minor issues that need further attention, including the numbering of atoms of nucleotides C1063 and C1066 in Fig. 2 (Spectinomycin) and the coloring scheme of a few figure panels (see comments).

Original comment

Fig. 2 Spectinomycin

In the text on page 8, "...where two waters are coordinated by the oxygens (O7) located in the nucleobases of C1063 and C1066..." Based on the figure, it appears it is the oxygen O2 in C1063 and C1066 that interacts with the waters.

Authors' response

Both the O7 and O2 of C1063 and C1066 coordinate water molecules...this is seen in Figure 2 and stated in the text on page 8 "where two waters are coordinated by the oxygens (O7) located in the nucleobases of C1063 and C1066, together with their respective ribose 2' oxygens (Fig. 2)."

The authors should use the standard atom numbering in nucleotides. There is no oxygen O7 in the nucleobase of cytosine. The exocyclic oxygen in C and U is O2, while in the ribose, they are O2', O3', O4', and O5'. For reference, please see <https://www2.tulane.edu/~biochem/nolan/lectures/rna/frames/nucs.htm>

We have changed O7 to O2 on page as requested.

Original comment

Fig. 4

Many of the panels are overwhelming. One that is particularly difficult to grasp is 4g. Try alternate orientations?

Authors' response

Unfortunately, it is very difficult to find a single view where all water molecules are observed. We realize that 4g is rather complicated but it's the same view as 4h and 4i, and the only view where one can nicely see the overlap between the waters in these comparisons. An alternative view in 4g will disconnect it from 4h and 4i. Nevertheless, we have taken the reviewers advice from the comment below and with a reduced atom colouring focussing on the interactions, we hope that this makes the presentation less overwhelming.

Panel 4g is now clearer. I suggest to attempt a similar coloring scheme for panels 4a, 4d, and 4e. It would improve the clarity of the figure.

We have now used the same coloring scheme for panels 4a, d, e as requested

Similarly, using the same coloring strategy for the insets in the Ext. Data Fig. 5, in Ext. Data Fig. 8a, g, and j, and in Ext. Data Fig. 9a-b, d-e, g-h, j-k will also improve these figures.

We have now used the same coloring scheme for Ext. Data Fig. 5, in Ext. Data Fig. 8a, g, and j, and in Ext. Data Fig. 9a-b, d-e, g-h, j-k as requested

Final Decision Letter:

Message 26th Jun 2023

:

Dear Professor Wilson,

We are now happy to accept your revised paper "Structural conservation of antibiotic

interaction with ribosomes" for publication as a Article in Nature Structural & Molecular Biology.

Your paper will be published online soon after we receive proof corrections and will appear in print in the next available issue. You can find out your date of online publication by contacting the production team shortly after sending your proof corrections. Content is published online weekly on Mondays and Thursdays, and the embargo is set at 16:00 London time (GMT)/11:00 am US Eastern time (EST) on the day of publication. Now is the time to inform your Public Relations or Press Office about your paper, as they might be interested in promoting its publication. This will allow them time to prepare an accurate and satisfactory press release. Include your manuscript tracking number (NSMB-A47395B) and our journal name, which they will need when they contact our press office.

About one week before your paper is published online, we shall be distributing a press

release to news organizations worldwide, which may very well include details of your work. We are happy for your institution or funding agency to prepare its own press release, but it must mention the embargo date and Nature Structural & Molecular Biology. If you or your Press Office have any enquiries in the meantime, please contact press@nature.com.

Please note that *Nature Structural & Molecular Biology* is a Transformative Journal (TJ). Authors may publish their research with us through the traditional subscription access route or make their paper immediately open access through payment of an article-processing charge (APC). Authors will not be required to make a final decision about access to their article until it has been accepted. Find out more about Transformative Journals <https://www.springernature.com/gp/open-research/transformative-journals>

Authors may need to take specific actions to achieve [compliance with funder and institutional open access mandates](https://www.springernature.com/gp/open-research/funding/policy-compliance-faqs). If your research is supported by a funder that requires immediate open access (e.g. according to [Plan S principles](https://www.springernature.com/gp/open-research/plan-s-compliance)) then you should select the gold OA route, and we will direct you to the compliant route where possible. For authors selecting the subscription publication route, the journal's standard licensing terms will need to be accepted, including [self-archiving policies](https://www.springernature.com/gp/open-research/policies/journal-policies). Those licensing terms will supersede any other terms that the author or any third party may assert apply to any version of the manuscript.

Sincerely,

Dimitris Typas
Associate Editor
Nature Structural & Molecular Biology
ORCID: 0000-0002-8737-1319